# Thawing permafrost poses environmental threat to thousands of sites with legacy industrial contamination

Moritz Langer [1,2] ✉, Thomas Schneider von Deimling [1,3], Sebastian Westermann[4,5], Rebecca Rolph[1,3], Ralph Rutte [6], Sofia Antonova[1], Volker Rachold[7], Michael Schultz [8], Alexander Oehme [1,3] & Guido Grosse [1,9]

Industrial contaminants accumulated in Arctic permafrost regions have been largely neglected in existing climate impact analyses. Here we identify about 4500 industrial sites where potentially hazardous substances are actively handled or stored in the permafrost-dominated regions of the Arctic. Furthermore, we estimate that between 13,000 and 20,000 contaminated sites are related to these industrial sites. Ongoing climate warming will increase the risk of contamination and mobilization of toxic substances since about 1100 industrial sites and 3500 to 5200 contaminated sites located in regions of stable permafrost will start to thaw before the end of this century. This poses a serious environmental threat, which is exacerbated by climate change in the near future. To avoid future environmental hazards, reliable long-term planning strategies for industrial and contaminated sites are needed that take into account the impacts of cimate change.

In the Arctic permafrost region, near-surface air temperatures are rising at rates at least two times faster than the rest of the globe[1,2], with latest data analyses suggesting up to four-fold faster warming[3], substantially changing the ground stability and hydrological conditions[4,5]. A recent review highlighted the potential of new biogeochemical risks that could be associated with permafrost thaw and the mobilization of hazardous substances from various sources[6]. At the same time, there is clear evidence of increased risk to the stability of infrastructure in permafrost regions[7–9], which was built on the premise of permanently frozen ground. One prominent environmental disaster, attributed in part to the loss of soil stability[10], was the spillage of 17,000 tons of diesel fuel from a destabilized tank facility near the industrial city of Norilsk in northern Siberia in May 2020, which entered the Arctic ecosystem and contaminated rivers, lakes, and tundra in a large permafrost watershed.

For decades, industrial and economic development of the Arctic was based on the assumption that permafrost would serve as a permanent and stable platform[11]. Past industrial practices also assumed that perennially frozen ground would function as long-term containment for solid and liquid industrial waste due to its properties as a hydrological barrier[12–14]. These widespread practices across the Arctic led to the accumulation of various toxic substances on or in permafrost. Known industrial waste types include drilling and mining wastes, toxic substances like drilling muds and fluids, mine waste heaps, heavy metals, spilled fuels, and radioactive waste (Fig. 1 Supplementary Table 1). Scientifically documented methods for dealing with such substances in remote Arctic regions during much of the 20th century include creating covered waste dumps in permafrost, covered drilling mud sumps, using hydrologically closed lakes and basins as natural dumps, and spreading substances across a large area for dilution in the belief that permafrost beneath and in the surrounding terrain would serve as a stable waste containment barrier of infinite duration[15,16]. A number of experiments were conducted in Alaska,

[1]Permafrost Research Section, Alfred Wegener Institute Helmholtz Centre for Polar and Marine Research, Potsdam, Germany. [2]Department of Earth Sciences, Vrije Universiteit Amsterdam, Amsterdam, The Netherlands. [3]Geography Department, Humboldt-Universität zu Berlin, Berlin, Germany. [4]Department of Geosciences, University of Oslo, Oslo, Norway. [5]Centre for Biogeochemistry in the Anthropocene, University of Oslo, Oslo, Norway. [6]Freelancer, Eppstein, Germany. [7]German Arctic Office, Alfred Wegener Institute Helmholtz Centre for Polar and Marine Research, Potsdam, Germany. [8]GIScience, Heidelberg University, Heidelberg, Germany. [9]Institute of Geosciences, University of Potsdam, Potsdam, Germany. ✉e-mail: moritz.langer@awi.de

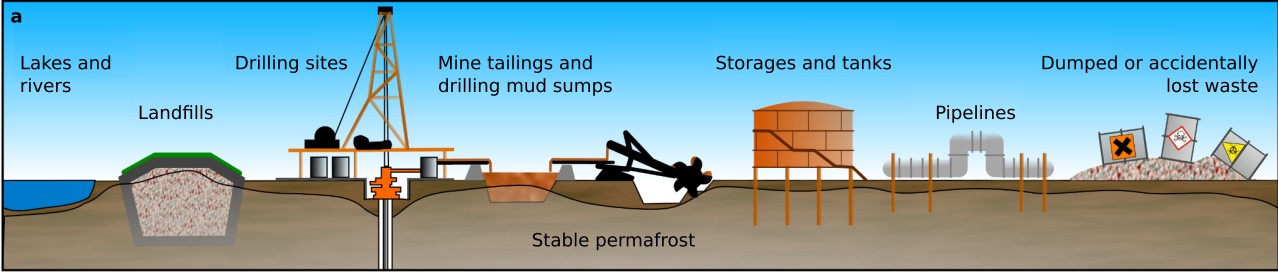

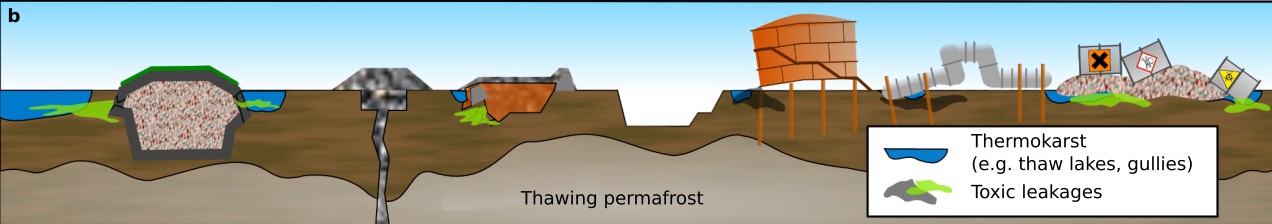

**Fig. 1 | Potential impacts of thawing permafrost on above- and below-ground industrial infrastructure containing toxic substances or waste.** Past and present industrial activities in the Arctic result in various accumulations of hazardous substances in the Arctic **a**. The warming and thaw of near surface permafrost unlocks frozen disposal sites and destabilizes foundations and containment structures **b**. Furthermore, permafrost thaw intensifies thermo-hydrological erosion and increases the lateral flow of water, fostering the dispersion of contaminants.

Canada, and Russia in which toxic liquids and solids, including radioactive waste, were deliberately placed in permafrost for containment[17–19]. To date, there has been no assessment of the environmental impact of these activities on the Arctic as a whole. The thawing of permafrost makes such assessments all the more urgent as the potential risk of toxic release and mobilization continues to increase. In addition, it is foreseeable that thawing permafrost will substantially increase the cost of mitigation and adaptation measures, including maintenance, replacement, relocation of infrastructure, and remediation of contaminated sites[7,8,20,21].

In this study, we present a pan-Arctic compilation of the number of industrial sites in permafrost-dominated regions and estimate the number of contaminated sites associated with these sites. We use the Northern Hemisphere Permafrost Map[22] to delineate the permafrost model domain, considering only areas with a permafrost occurrence probability of over 50%. Based on data from Alaska, we also assess the different types of toxic substances associated with industrial activities in these regions. We also use future climate scenarios to predict how many industrial and contaminated sites will begin to be affected by permafrost thaw. Our results suggest that industrial legacy sites in thawing permafrost and the release of toxic substances pose a significant environmental risk at the pan-Arctic scale, requiring management strategies.

## Results

### Industrial sites in permafrost dominated Arctic regions

We synthesized a geospatial dataset of industrial sites based on OpenStreetMap (OSM) and the Atlas of Population, Society and Economy in the Arctic (APSEA) to portray the spatial pattern of industrial activities in the permafrost model domain. This shows that about 4500 land use elements labeled industrial (hereafter called industrial sites) are located within the permafrost dominated regions of the Arctic (Fig. 2). Following the Intergovernmental Panel on Climate Change (IPCC) classification scheme for industry sectors (Agriculture, Forestry and Other Land Use (AFOLU); Energy; Industrial Processes and Product Use (IPPU); Waste), we find that among the clearly labeled data, the classes Energy and Agriculture, Forestry and Other Land Use are the largest (each >10%). However, most industrial sites (>65%) within the OSM-APSEA database are either not clearly labeled or cannot be assigned to one of the four major IPCC classes and are, thus,

considered uncertain. This incomplete labeling leads to large uncertainties in quantifying specific industrial sectors, but the dataset nevertheless provides a comprehensive and consistent assessment of the spatial distribution of industrial sites for the investigated permafrost model domain. We relate this spatial distribution of industrial sites to the occurrence of contaminated sites in the following assessment.

### Region specific characteristics of contaminated sites

In order to gain a better understanding of the relationship between industrial sites and the occurrence of contamination, we use regional data available for Alaska (Contaminated Sites Program, CSP[23]) and Canada (Federal Contaminated Sites Inventory, FCSI[24]). The synthesized dataset (CSP/FCSI) allows us to quantify the extent and nature of contamination and its spatial relationship to industrial sites (Fig. 3) for the permafrost dominated regions on the North American continent. We then extrapolate the spatial relationship between industrial sites and contaminated sites in the North American dataset to the Arctic as a whole, and validate the pan-Arctic dataset with a compilation of Russian sites based on publicly available sources (see below).

The CSP/FCSI dataset as of January 2021 shows about 8000 individual contaminated sites for Alaska and about 22,000 contaminated sites for Canada. About 18% (~3600) of these sites are located in permafrost dominated areas. Approximately 30% of the surveyed contaminated sites in the investigated permafrost domain of the North American continent are considered active, meaning that cleanup has not yet been completed or the status of the contamination has yet to be clarified. While the CSP and FCSI data provide relatively consistent information on site and treatment status, the two databases differ greatly in terms of information on industrial origin, historical record, and financial costs associated with site management. Therefore, we combine the two datasets only to analyze the spatial correlation between industrial sites and the occurrence of contaminated sites. A detailed analysis of toxic substances and their industrial origins focuses exclusively on the Alaskan data.

Within the Alaskan CSP dataset more than 60% (~850) of the contaminated sites located in the permafrost dominated regions are explicitly associated with industrial or military activity[23]. The CSP database further reveals that the annual number of newly registered contaminated sites peaked in the 1990s and decreased from about 90

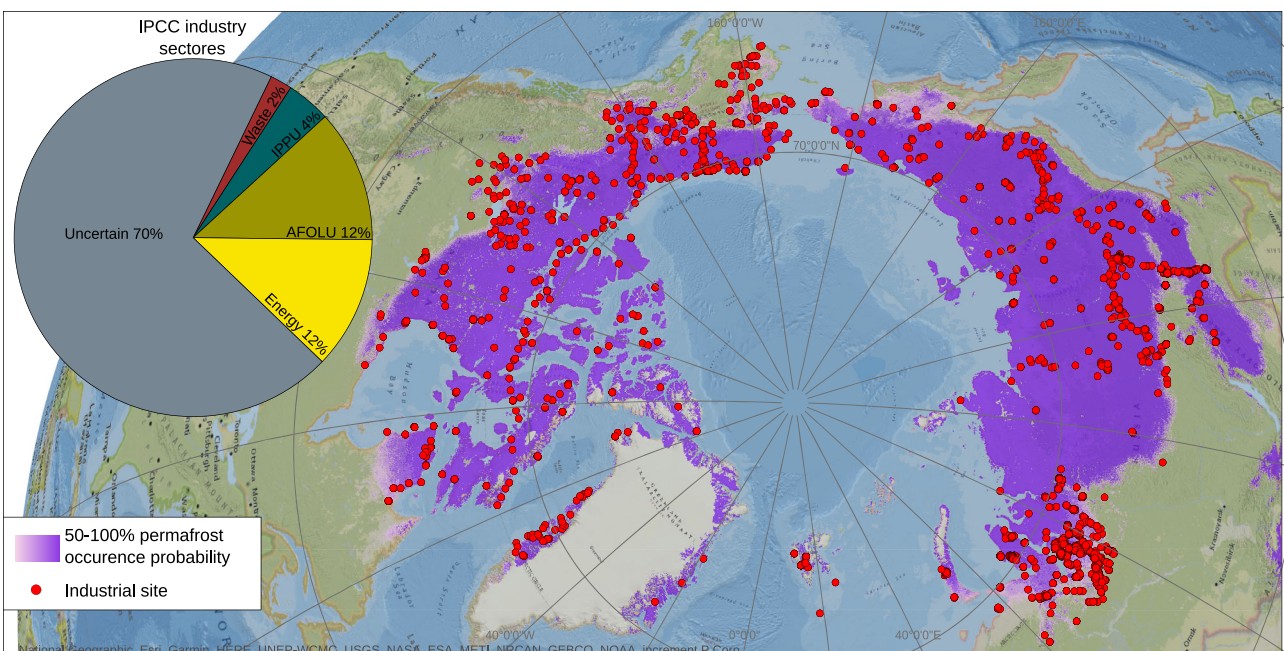

**Fig. 2 | Occurrence of industrial sites in permafrost dominated regions of the Arctic.** The permafrost model domain (permafrost occurrence probability >50%) is delineated based on the Northern Hemisphere Permafrost Map (NHPM)[22] and the database of industrial sites is based on OpenStreetMap (OSM) and the 2019 Nordregio Atlas of Population, Society and Economy in the Arctic (APSEA). Background map is based on the National Geographic World Map[58]. While the industrial sectors Energy and Agriculture, Forestry and Other Land Use (AFOLU) account for the largest proportion of industrial sites among the clearly labeled data, more than 65% of the mapped industrial sites are not clearly labeled. This creates a large uncertainty in quantifying specific industry sectors and highlights the need for improved databases on industrial activities in the Arctic. Maps were created by using ArcGIS version 10.5 (Esri Inc., USA).

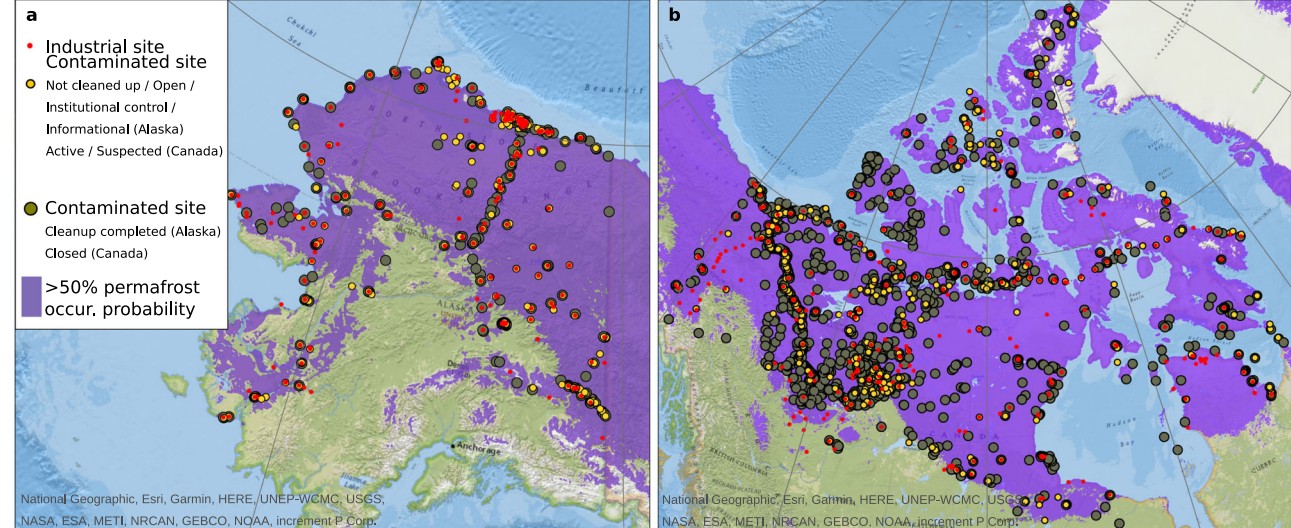

**Fig. 3 | Industrial and contaminated sites within permafrost dominated regions of the North American continent.** Maps show the distributions of industrial and contaminated sites located within permafrost dominated regions[22] in **a** Alaska as reported by the Contaminated Sites Program (CSP)[23] and **b** in Canada as reported by the Federal Contaminated Sites Inventory (FCSI)[24]. Background map is based on the National Geographic World Map[58]. Our classification Cleanup complete and Not cleaned up refers to information on whether hazardous substances have been removed or remain (or are suspected to remain) in the environment. Maps were created by using ArcGIS version 10.5 (Esri Inc., USA).

registrations per year in 1992 to 38 in 2019, roughly following the pattern of crude oil production from the Alaskan North Slope (Supplementary Fig. 1). This example clearly shows that 20th century industrial development in Alaska led to an accumulation of contaminants that often remained at the industrial sites.

Based on the absolute occurrence of industrial sectors in the CSP data (Supplementary Fig. 2) we expect a large fraction of the existing contaminations to originate from the Industrial Processes and Product Use (~30%) and Energy (~25%) sectors[23]. These two sectors are responsible for more than 50% of the contaminations, but together account for only about 16% of the industrial sites in the Alaskan permafrost model domain. The CSP data reveal that fuels such as diesel, kerosene, gasoline and associated chemicals, constitute by far the largest fraction (about half) of the substances found at contaminated

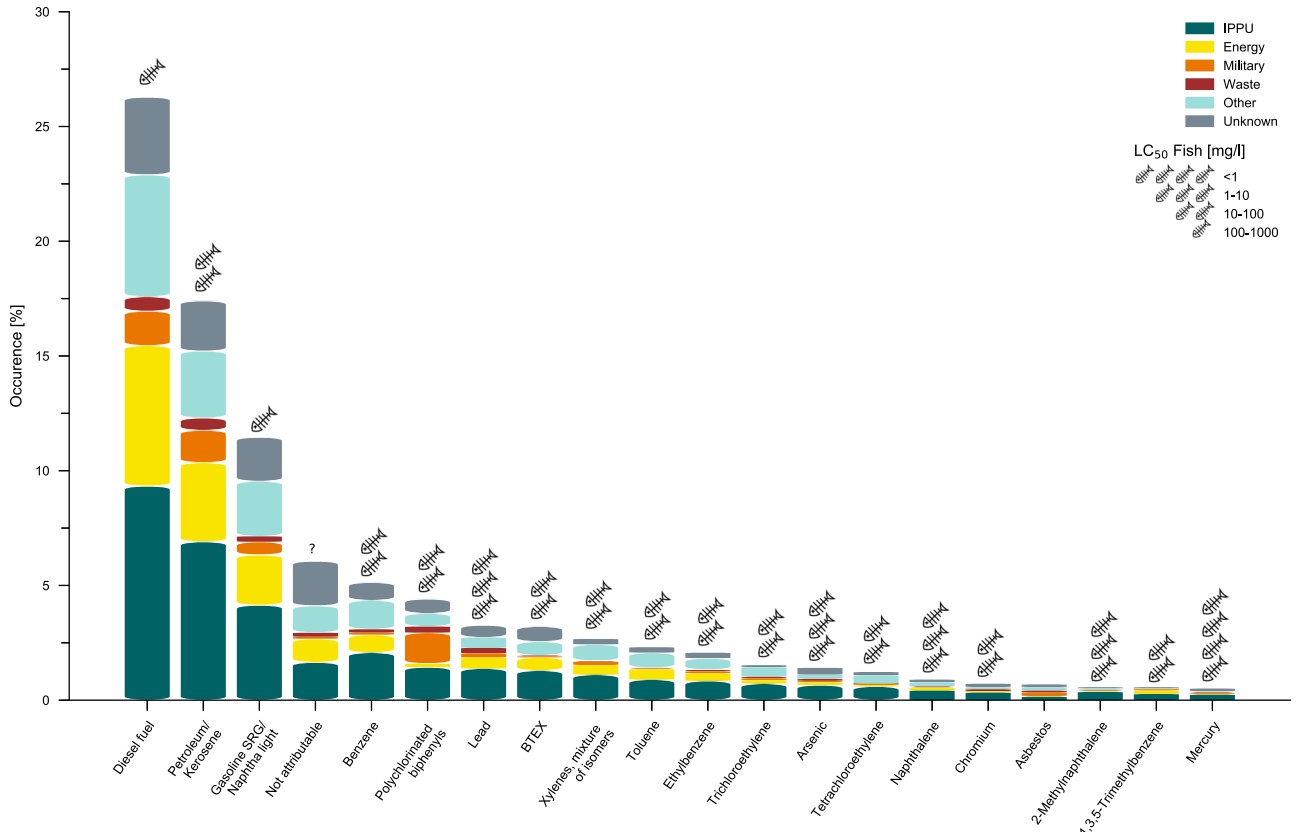

**Fig. 4 | Toxic substances at contaminated sites in the permafrost dominated regions of Alaska.** The stacked bar plot depicts the relative occurrence of the most common toxic substances found at contaminated sites in permafrost dominated regions of Alaska (based on 22), as reported by the Contaminated Sites Program (CSP). The occurrence is further differentiated by industry sector: Industrial

Processes and Product Use (IPPU), Energy, Military, and Waste. Toxic substances from Agriculture, Forestry, and Other Land Use (AFOLU) occur in negligible numbers. The toxicity of each substance is indicated using the median lethal concentration for fish (LC$_{50}$-fish) after 96 h (see also Supplementary Table 1).

sites in the Alaskan permafrost model domain (Fig. 4). Substances with high aquatic toxicities such as mercury, lead, and arsenic are also amongst the top twenty of the most frequently listed substances in the CSP database. We emphasize that about 40% of the industrial sites, about 20% of the contaminated sites, and about 10% of the substances are labeled as unknown or not attributable in the database so that any relational analysis between these categories would be highly uncertain. However, our analysis clearly shows the magnitude and general properties of contaminated sites in Alaskan permafrost, highlighting the large amounts of hazardous substances that remain as a legacy of industrial activities.

## Pan-Arctic occurrence of contaminated sites in permafrost dominated regions

We performed a spatial cross-analysis between the locations of industrial sites and contaminated sites within the permafrost dominated regions of Alaska and Canada (Fig. 5a, b) which reveals that the location of industrial sites can provide a first-order estimate on where dangerous substances can be expected (Supplementary Fig. 3). Two point process models fitted to the CSP/FCSI data provide the upper and lower bounds of the observed relationship (Supplementary Fig. 4). To assess the validity of the extrapolated dataset for Russia (where data similar to CSP/FCSI are lacking), we compiled a database of industrial contamination incidents based on Russian media and other publicly available sources (from 2000 until 2022, see Methods). This database contains 44 individual incidents located within the permafrost dominated region and 58 incidents outside this region (Fig. 5c). Although our database contains only a small subset of the actual contaminated

sites in Russia, the recorded incidents provide clear evidence that the spatial relationship between industrial sites and contaminated sites also applies to the Russian permafrost dominated region (Supplementary Fig. 5). We conclude that it is reasonable to estimate the number of contaminated sites in the permafrost model domain from existing maps of industrial sites for at least the Alaskan, Canadian, and Russian sectors of the Arctic, which together comprise more than 98% of the recorded industrial sites. As a consequence, we use the models to estimate the total number of industrially contaminated sites for the entire Arctic permafrost model domain (Fig. 5d).

The extrapolations yield total estimates of 13,000 to 20,000 contaminated sites which potentially exist in the permafrost dominated regions in the Arctic (Table 1). Most of these sites (70.5 ± 5.5%) occur in Russia, which also hosts the largest percentage (~85%) of industrial sites. Canada and Alaska together host the second largest percentage (~18%) of industrial sites and the second largest share (23 ± 5%) of contaminated sites in the permafrost model domain. Greenland and Svalbard are estimated to have only a small percentage (<5%) of industrial and contaminated sites within permafrost dominated regions. We emphasize that these estimates of the absolute numbers of contaminated sites have large uncertainties, especially for Russia. There, the two point process models differ in the predicted number of contaminated sites by a factor 1.7. The occurrence of contaminations in the Arctic shows strong regional differences that likely depend on the industrial sectors operating in these regions, national environmental legislation, and its enforcement over time. In the absence of comprehensive data for Russia, our upscaling approach does not capture these dependencies, resulting in the large

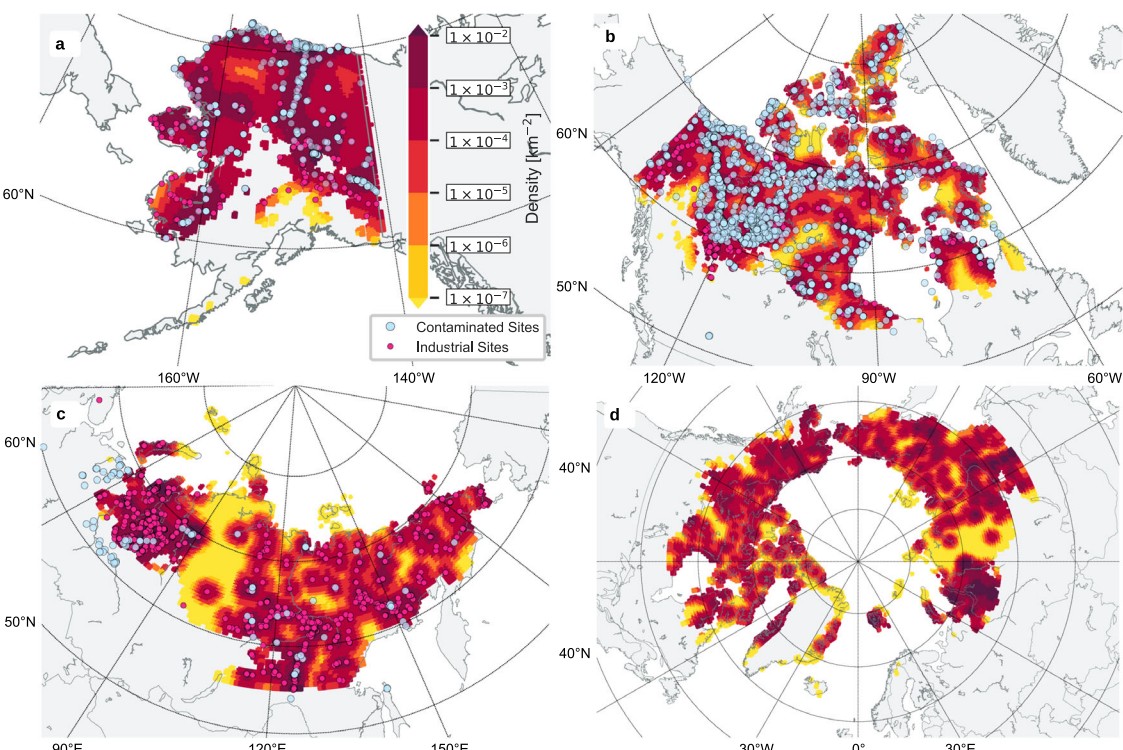

**Fig. 5 | Projected densities of contaminated sites in Arctic permafrost dominated regions.** The density maps of contaminated sites per area as derived by the point process models for **a** Alaska and **b** Canada with dots depicting the locations of industrial and contaminated sites. Map **c** shows the predicted density of contaminated sites for the permafrost dominated region in Russia with dots showing the data on contaminated sites surveyed in this study and that are used for model validation. Applying the model to the entire permafrost model domain yields a pan-Arctic map of estimated contaminated site density **d**. Note that the speckled appearance results from the regional clustering of industrial sites combined with the chosen bandwidth ($50 \times 50$ km) of the gaussian density filter used for the point process models. Map generated with Python using the Basemap Matplotlib library and the GSHHG dataset[59].

**Table 1 | Numbers of industrial and contaminated sites within the permafrost dominated region of the Arctic**

| Country | Number of industrial sites within the model domain (OSM-APSEA) | Number of contaminated sites in permafrost model domain | Estimated range of contaminated sites in permafrost model domain (PPM1 - PPM2) |
|---|---|---|---|
| Russia | 3909 | Own data collection through media research | 8538–15149 |
| Canada | 484 | 2502 | 2308–2720 |
| Alaska | 343 | 1148 | 1186–1493 |
| Greenland | 46 | No data | 296–301 |
| Svalbard | 30 | No data | 306–681 |
| Total | 4494 | No data | 13047–19933 |

The estimated numbers of contaminated sites are based on two point process models (PPM1 and PPM2). The two models represent the upper and lower bounds of the observed relationships between the spatial densities of industrial sites and contaminated sites.

uncertainties in our estimate. A more detailed analysis of the contamination potential would require (i) a dataset similar to CSP and FCSI for the entire Arctic and (ii) a characterization and quantification of contaminant stocks including storage as well as intentional or accidental disposal, covering the entire range from small fuel spills to major contamination of the magnitude of the 2020 Norilsk diesel spill.

**Industrial and contaminated sites affected by permafrost thaw**
We applied a numerical permafrost model (see Methods), driven by past climate data and future projections, to quantify how current and future permafrost thaw may affect industrial and contaminated sites in the permafrost model domain (Fig. 6a). The model has a coarse spatial resolution (1 degree), and thawing of permafrost within grid cells is considered likely if simulations indicate the formation of a persistent talik (defined here as a permanent unfrozen soil layer greater than 0.1 m thick above permafrost). The simulations predict that about 22%

of the currently existing industrial sites (~1000) and $20 \pm 4\%$ of the estimated contaminated sites (2200–4800) are located in a region within which the simulations indicate that permafrost degradation is possible under present climate conditions (2020). The simulations further indicate that 15% of these industrial and contaminated sites are located in grid cells where a change from stable (talik free in the model) to unstable permafrost (talik in the model) occurred between 1960 and 2000. An accelerated increase of sites located in the modeled zone of degrading permafrost between 2000 and 2020 is due to a spatial clustering of industrial sites in the southern zone of the permafrost model domain (Supplementary Fig. 3). This zone was simulated to be affected by permafrost thaw when global warming exceeded about 0.5 °C above pre-industrial conditions (1850–1900) (Fig. 6b). It is likely that many of the industrial sites in this zone were built at permafrost-free locations where this was possible or where permafrost soil had been actively removed prior to construction, so

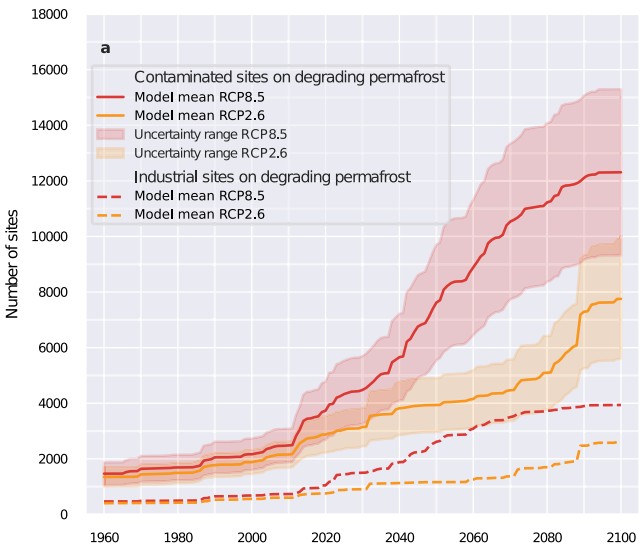

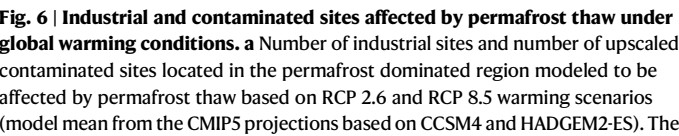

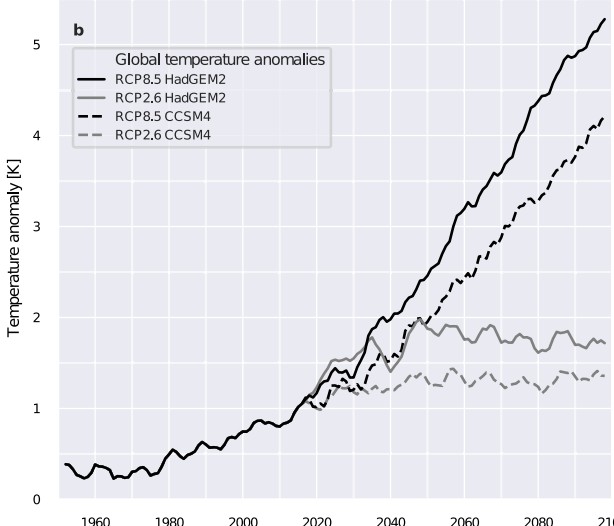

**Fig. 6 | Industrial and contaminated sites affected by permafrost thaw under global warming conditions. a** Number of industrial sites and number of upscaled contaminated sites located in the permafrost dominated region modeled to be affected by permafrost thaw based on RCP 2.6 and RCP 8.5 warming scenarios (model mean from the CMIP5 projections based on CCSM4 and HADGEM2-ES). The shaded areas show the uncertainty range due to the spatial extrapolations based on the two point process models (PPM1 and PPM2). **b** The related global temperature increase is shown as an anomaly compared to the pre-industrial period (1850–1900).

that further warming would not affect their stability. However, at some of these sites, it cannot be ruled out that permafrost and possibly ground ice may still be present and that further warming will lead to soil instability and the formation of new hydrologic pathways. At such sites, the risk of destabilization and contaminant mobilization due to permafrost degradation is largely unpredictable without detailed knowledge of the local ground stratigraphy and infrastructure design.

About 77% of the existing industrial sites (~3500) and 60 ± 15% of the estimated contaminated sites (5800–15,100) are located in regions simulated to be stable (talik free in the model) under current climate conditions (2020). The projections suggest that these sites will decrease in number by about 3 ± 2% for a low (RCP 2.6) and 46 ± 13% for a high (RCP 8.5) greenhouse gas emission scenario within the next 30 years until 2050. After 2050, the number of industrial sites located in regions affected by permafrost degradation is estimated to increase by 1100 from 2020 to 2100 under the low emissions scenario consistent with the 2 °C global warming target (RCP 2.6). For contaminated sites, simulations under RCP 2.6 estimated an increase of between 3,400 and 5,200 sites located in regions affected by permafrost thaw. In contrast, almost all existing industrial and contaminated sites will be located in regions affected by permafrost degradation under a business as usual scenario (RCP 8.5). This strong warming scenario results in widespread near-surface permafrost thaw in the simulations, affecting almost the entire Arctic by 2100 as predicted by other modeling studies[25]. Our findings underline that considering timeframes beyond 2050 is essential for assessing risks related to failure of industrial infrastructure and thaw-mobilization of contaminants—a time horizon neglected in recent studies[7,8], infrastructure planning and the management of contaminations.

The projections of permafrost degradation underline the problem of stability loss at most existing industrial sites and their risk for potential release of toxic legacies (contaminated sites). This is especially true for sites in operation before the 2000s, since thermal permafrost conditions have already changed substantially over the last two decades[26].

Our simulations underscore the need to extend the time horizon of risk studies, which have so far focused on the middle of the century[7]. Risk studies need to consider the time scale of permafrost dynamics: the consequences of past and present permafrost degradation will extend into the second half of the 21st century and beyond, even if climate warming is halted. Our results indicate that staying below the 2 °C global warming target may avoid a further rapid increase in the number of affected industrial sites and contaminated sites in the second half of the 21st century. The pronounced difference in the number of industrial sites affected by permafrost degradation between the two scenarios is mainly due to the specific latitudinal distribution of the industrial sites (Supplementary Fig. 3).

Our estimate of the increase in the number of affected sites is likely conservative because it does not account for rapid permafrost thaw processes (e.g. thermokarst and thermal erosion)[4], the effects of infrastructure on permafrost thaw[26], or the effects of contamination (e.g. freezing point depression from dissolved substances)[27] on permafrost stability.

## Discussion

Our analysis suggests that there are currently thousands of industrial sites located on degrading permafrost, which are exposed to the risk of partial or complete failure. Climate warming will increasingly aggravate the probability that toxic substances associated with such sites will be released due to the failure of critical facilities for waste, storage, and disposal[7]. We emphasize that permafrost already loses substantial bearing capacity at temperatures close to 0 °C, so that destabilization of any infrastructure may occur even if permafrost is assumed to be stable according to our simulation. This aspect is further underlined when considering that permafrost degradation can be strongly enhanced through the impact of infrastructure on ground thermal conditions which is not accounted for in this study. Permafrost thaw not only poses a risk for industrial infrastructure integrity and disposal areas, but will also open new hydrological pathways[5,11,28] and reduce the accessibility of remote permafrost regions[29]. The thaw-induced changes in water flow pathways and hydrologic connectivity will impact the release and dispersal of contaminants, while loss of ground stability will limit access to impacted sites and use of heavy equipment, both severely complicating mitigation and cleanup measures. Depending on the type of contamination, high ion concentrations can occur in the pore water and lead to an unfrozen cryotic zone where substantial groundwater movement can occur even before the permafrost thaws.

The Contaminated Sites Program (CSP) in Alaska and the Federal Contaminated Sites Inventory (FCSI) in Canada demonstrate the value of transparent documentation of contamination status for science-based assessments. Without this comprehensive documentation, covering approximately 40% of the permafrost dominated region of the Arctic, our large-scale assessment of contaminated sites would not have been possible. While our analysis provides an initial estimate of the magnitude of the environmental risk, it also demonstrates the large uncertainty that results from incomplete data. We were able to reduce the data gap partly by using public media sources to compile a geospatial data subset on contaminated sites for the Russian Arctic. However, we note that there is still a great lack of public information on industrial sites and activities in much of the Arctic preventing a more comprehensive risk assessment. Currently available data are insufficient to assess the likelihood of future contamination or the risk of mobilization of contaminants due to permafrost degradation. This would require more detailed information on the quantity and toxicity of industrially used substances, on their leachability potential (water solubility), proximity to hydrological pathways, conditions of storage or disposal, and site-specific environmental conditions such as permafrost and ground stratigraphy. In particular, there is a lack of transparent documentation of environmentally hazardous substances transported to and used in the Arctic[30]. Moreover, there are no international environmental regulations for the Arctic, as formulated for the Antarctic in the Madrid Protocol, that require transparent documentation of contamination and potential sources of hazardous substances[31]. At the same time, the lack of information is exacerbated by a rapidly growing number of industrial sites due to increasing economic interest and development in the Arctic[32].

Whereas risks associated with municipal waste and how that is affected by climate change has received attention[33,34], the risks of contamination from large-scale and gradual release of hazardous substances as a consequence of permafrost thaw are still poorly studied. Consistent with studies predicting the costs of infrastructure destabilization due to permafrost thaw[8,9], the costs of securing contaminants, post-operational renaturation, and remediation following an accidental release of contaminants can be expected to increase sharply. Consideration of these financial risks likely modifies the cost-benefit calculations for industrial activities in the Arctic.

Recently,[35] suggested mercury release as a potential threat from thawing permafrost, and[6] generally discusses bio-geochemical risks from permafrost thaw. We extend this discussion by considering risks and consequences of potential abrupt contamination from failure of industrial infrastructure built on permafrost (e.g., as happened for the Norilsk diesel tank incident). This type of risk shows a clear threshold behavior associated with reaching a critical ground temperature[36]. The risk of abrupt contaminant release could potentially be managed by reducing site-level vulnerability. However, this would require operational monitoring systems to deliver the early warnings needed for timely infrastructure adaptation. Such technology-based mitigation strategies could be fairly straightforward to apply to new industrial facilities and to sites currently in operation, but seamless monitoring and remediation of legacy industrial sites appears rather unrealistic. Only few monitoring programs exist for legacy sites, such as for the covered drilling waste sumps at the Mallik drill site in the Mackenzie Delta of NW Canada; however, the observation period (7 years in this particular example) clearly needs to be longer to capture the time scales of thawing permafrost in the coming decades[37]. In this context, it is also important to emphasize that permafrost thaw usually affects larger regions so that there is a risk of multiple contaminated sites leaking at the same time. Therefore, technologies such as remote sensing, modeling, or selective monitoring of sites representative of larger regions could provide early warning for legacy sites. Consequently, we suggest that for contaminated industrial sites on permafrost, it is particularly important to reconsider the time, effort, and financial resources required for official long-term monitoring and site remediation. This includes the development of post-thaw scenarios. For most existing sites in the Arctic, the time period to consider may extend well into the second half of the 21st century and beyond due to sustained permafrost thaw. Our initial assessment suggests that existing and old industrial sites in the Arctic will be increasingly affected by climate change. The effects of thawing permafrost, with all its consequences such as loss of hydrological barriers, improved hydrological connectivity, reduced soil stability and strongly impeded site accessibility for clean-up measures, will often occur after the operating period of industrial sites. This underscores the need to avoid leaving environmentally hazardous substances at the sites, as permafrost can no longer be considered a reliable barrier to their containment. Furthermore, long-term remediation strategies will be necessary for contaminated legacy sites that have already been closed if they still contain hazardous substances.

## Methods

### Geospatial data on industrial sites

The total number of industrial sites located within the permafrost dominated region of the Arctic was estimated using OpenStreetMap (OSM) in combination with a spatial dataset on industrial facilities from the Atlas of Population, Society and Economy in the Arctic (APSEA) provided by the Nordregio. OSM is a thematic, limitless open spatial database under continuous development with thousands of everyday modifications[38]. Given the application of certain filters, specific themed applications like land cover can be extracted using OSM[39]. For our study, we used all OSM objects that were either a node (point) or way (polygons only) and attributed with the key landuse or building and the value industrial north of 55° N latitude. A total of 90,018 areas, 108,427 buildings and 139 detailed facilities related to industrial activities were extracted.

The OSM data were complemented with the APSEA data providing additional data on industrial sites such as airports, ports, mines, pipelines, roads, and petroleum fields. To avoid double counting and resolve geometry conflicts between both datasets, we define buffer areas (1 km distance) around all features as overlap merging criteria.

Identification of the distribution of the combined OSM-APSEA database by major permafrost presence was achieved by overlay analysis using the Northern Hemisphere Permafrost Map (NHPM)[22]. Hence, sites in a grid cell characterized by a modeled permafrost occurrence probability greater than 50% were kept within the database, resulting in a total of 5,234 entries. Due to the spatial resolution of the NHPM (1 km), the occurrence of permafrost cannot be clearly assessed at the point locations of the industrial and contaminated sites. We therefore emphasize that we consider sites that are located in regions that are dominated by permafrost, but some of the actual sites may be permafrost free.

Semantic optimization of the database was done by a detailed word count analysis, removal of text duplicates and obsolete or non-relevant text characters. The database was summarized into four industrial sectors following the classification of the IPCC (Agriculture, Forestry and Other Land Use, Energy, Industrial Processes and Product Use, Waste). The majority of database entries remained unlabeled due to missing tags and were, thus, classified uncertain, a designation also used for tags not fitting into the other categories above. The resulting database enabled a first-order assessment of the spatial distribution and concentration of industrial sites. However, historic industrial sites such as fully or partially revegetated old drill pads and drilling sumps, covered landfills, and features as yet undetected by remote sensing such as individual borehole sites, might be absent from our database. Furthermore, our database does not account for pipeline networks transporting oil and gas which span thousands of kilometers across the Arctic. Completeness and availability of land description in OSM can vary temporally and spatially[40]. The completeness of the combined

OSM-APSEA database on industrial sites was assessed by comparison with the Arctic Coastal Human Impact (SACHI) dataset[32], which provides a map of infrastructure areas along Arctic shorelines impacted by permafrost (limited to 100 km inland). To compare the spatial consistency in terms of the occurrence of industrial sites between the two datasets, we allowed a buffer radius of 100 m around the OSM-APSEA point data. In this way, the existing positives from OSM-APSEA were evaluated for the entire SACHI domain (Supplementary Fig. 6). Missing positives were only evaluated for selected test regions (Alaska, Canada, Greenland, and Russia), as it was necessary to manually select the infrastructure areas within SACHI that represent compatible industrial sites. The assessment showed that the presence of industrial sites in the OSM-APSEA database is more than 85% consistent with the presence of industrial infrastructure in the SACHI database for the test regions and the entire SACHI domain. However, OSM-APSEA has a much lower number of industrial sites compared to SACHI ($40 \pm 20\%$ missing). The observed discrepancies are consistent with values from previous comparisons between SACHI and OSM[32]. Based on the database comparisons, we argue that OSM-APSEA provides a relatively reliable estimate of the regional presence of industrial sites in the Arctic, but that there is substantial uncertainty regarding the absolute number of individual infrastructure elements. Given this analysis and the fact that the above mentioned infrastructure elements were not included in the OSM-APSEA database, we assess our estimate of industrial sites as conservative.

### Geospatial data on contaminated sites

In order to investigate the link between industrial sites and the potential occurrence of contaminated sites we made use of datasets available for Alaska and Canada. For Alaska the Contaminated Sites Program (CSP) is published by the U.S. Department of Environmental Conservation[23] and for Canada the Federal Contaminated Sites Inventory (FCSI) is provided by the Treasury Board of Canada Secretariat.

Besides geographic coordinates, both datasets contain information on the treatment status indicating whether cleanup measures have been implemented. A site classification is provided for the CSP data which permitted a semantic analysis of the site names in order to classify the contaminated sites according to their industrial origin (Agriculture, Forestry and Other Land Use, Energy, Industrial Processes and Product Use, Waste). Site names not fitting into any of these categories are labeled as Others while missing site names are labeled as Uncertain (Supplementary Fig. 7). A hazard identification number in the CSP dataset provides access to web pages on which more detailed information is provided including a timeline of actions implemented as well as free-text site descriptions and chemical reports. For each contaminated site, the earliest date of the timeline was automatically extracted and used for further analysis. Frequencies of chemical substances listed in the CSP database were extracted semi-automatically using Levenshtein distances to identify keywords[41]. The extracted keywords were matched with substances listed in the database of the Chemical Abstracts Service (CAS) of the Chemical American Society[42]. Aquatic toxicities following the Globally Harmonized Systems (GHS) were extracted from the GESTIS substance database[43] as median lethal concentrations for fish ($LC_{50}$ Fish 96 h) and missing data were complemented by individual literature review.

### Spatial extrapolation of contaminated sites

In order to infer the probabilistic spatial relationship between the density of industrial sites and the occurrence of contamination, we set up a point process model assuming a nonhomogeneous poisson process. The method allowed us to derive a density map from the point dataset of industrial sites for the entire pan-Arctic permafrost model domain. Following extensive tests with bandwidths ranging between 10 km and 100 km, we used a gaussian kernel with a bandwidth set to 50 km which turned out to be a good compromise between the

preservation of spatial details and the number of sites considered. We tested two basic inhomogeneous poisson models which were fitted to CSP/FCSI point data filtered for industrial sites within the permafrost dominated region of Alaska and Canada using the point process modeling tools provided by R[44,45] (Supplementary Fig. 4). The fitted models are applied to the pan-Arctic industry density map in order to generate estimated intensity maps of contaminated sites (Fig. 5). We emphasize that we implicitly assume the data subset of the North American continent to be representative of the entire Arctic and that the relationship between industrial sites and contaminations follows an inhomogeneous Poisson process which is only partly correct according to the observed intensity function (Supplementary Fig. 4). We estimate the total number of potentially contaminated sites for the Arctic countries and the Arctic as a whole by integrating intensity maps within the different national boundaries (Table 1).

To validate the applicability of the point process models outside the North American continent, we created a sample dataset on industrial contamination in the permafrost region of Russia. The dataset was created based on a Google search using keywords in Russian (загрязнение−pollution; разлив нефтепродуктов−oil spill; техногенная авария−industrial accident or disaster; Арктика−Arctic; мерзлота−permafrost) and a search of several online media (including local and federal news portals). The database includes the locations of contamination incidents, the date, and information about the type of event, as well as the link to the media source. In some cases, the contaminated area and volume of the spill are also provided. However, we emphasize that it is generally impossible to independently verify these numbers. In order to avoid a bias due to expectation, the data search was performed without considering the map of projected contaminated sites. In total we obtained 102 incidents, starting mostly from the 2000s. As the search included either a general term permafrost or Arctic, only 44 incidents fell into the zones dominated by permafrost (occurrence probability greater than 50%), leaving another 58 incidents in the zones where permafrost occurrence is less likely. We analyzed their location in relation to the modeled intensity maps for Russia (Fig. 5c). To this end, we applied logarithmic binning to spatially summarize the model results into six intensity classes. For each intensity class, we calculated the expected total number of contaminated sites by spatial integration. We drew ($N = 1000$) random samples from the modeled total distribution, each comprising the number of observed contaminated sites ($n = 44$). The resulting distributions for each intensity class were then compared to the observed contaminated sites and their occurrence within modeled intensity class (Supplementary Fig. 5). The model-based random selection reproduces the observed distribution very well. The observed number of contaminated sites across all intensity classes is always within the plausible range of the predicted number of contaminated sites. This suggests that the spatial relationship between industrial sites and contaminated sites found for the permafrost region of the North American continent also applies to the Russian permafrost region.

### Projections of permafrost stability

For evaluating the future development of the thermal state of permafrost, we used the one-dimensional, transient permafrost model CryoGridLite, forced only by near surface air temperature and precipitation[46]. The applied climate forcing was based on ERA-Interim reanalysis data for simulating the thermal state of permafrost under historic and current climatic conditions, while future simulations were based on decadal monthly anomalies from the CCSM4[47] and HADGEM2-ES[48] CMIP5 projections under RCP 2.6 and RCP 8.5 scenarios. The scenarios used span a wide range of projected warming of the terrestrial Arctic (excluding glaciated regions) ranging from 0.7 °C to 8.0 °C over a period from 2020 to 2100. A long term model spin-up from 500 A.D. to 1979 was performed based on anomalies from paleoclimate simulations with the Mk3L climate model of the

Commonwealth Scientific and Industrial Research Organisation (CSIRO)[49]. Uncertainty in the future climate scenarios is considered by including 2 CMIP5 models of comparatively low (CCSM4[47]) and high (HADGEM2-ES[48]) projected Arctic warming which were run under low (RCP 2.6) and high (RCP 8.5) emission scenarios. The full spread of CMIP5 model projections is likely larger than our model range indicated in Fig. 6b. Furthermore, we note that the simulations did not take into account rapid thaw processes[50] which may be triggered by thawing excess ground ice and erosion[4,36].

Data required to set ground stratigraphies were extracted from the Open-ECOCLIMAP global database of land surface parameters[51,52]. Soil organic carbon contents were extracted from the Northern Circumpolar Soil Carbon Database version 2 (NCSCDv2)[53] and the thickness of the soil layer was determined according to the Gridded Global Data Set of Soil Thickness[54]. We emphasize that the spatial domain to which the CryoGridLite model was applied is defined by the extent of NCSCDv2 that covers the northern circumpolar permafrost region. Stratigraphies of soil physical parameters were derived using the parameterizations developed for the SURFEX land surface and ocean scheme[55]. Water or ice saturation of the ground was adjusted according to a global water table product[56] with soil layers below the water table assumed saturated, and soil layers above the water table set halfway between field capacity and porosity. Snow cover densities and compaction rates were parameterized with a bulk approach according to pan-Arctic climate classes[57]. Since the available parameter and forcing datasets represent regional averages rather than specific locations and the high computational demand of the simulations, we applied a coarse grid cell resolution of one degree. The modeling should be considered as a first-order assessment of the average thermal state of the subsurface within the respective model grid cells, which do not represent permafrost conditions at specific locations. Model code, parameter settings, and forcing data are accessible through Zenodo (10.5281/zenodo.7579177).

A strong indicator for permafrost degradation is the formation of a persistent year-round unfrozen layer above the permafrost, referred to as a talik. The simulations allow us to determine the date at which a persistent talik forms, which means that it does not occasionally refreeze in subsequent years. We define a talik to exist if a soil layer at least 0.10 m thick remains unfrozen. This criterion only applies to sites with permafrost, which means the profile of the annual maximum temperature has temperatures below 0 °C.

To evaluate the consistency between the NHPM[22] delineated permafrost model region and the CryoGridLite permafrost simulations, we compare the NHPM permafrost occurrence probabilities with the simulated presence or absence of persistent taliks (Supplementary Fig. 8). Regions with taliks and permafrost free areas were combined for this purpose. The comparison was made for the same reference period (2000–2016) and the results show that the majority of grid cells (80%) with simulated talik occurrence are located in regions where the probability of permafrost occurrence is low (less than 50%), while the majority of grid cells (80%) with simulated talik-free conditions are located in regions where the probability of permafrost occurrence is high (more than 90%). These limits align well with the values used to distinguish zones of continuous and discontinuous permafrost within the NHPM.

## Data availability
The geospatial data on industrial and contaminated sites, the model forcing data, and model parameters, are available from the Zenodo repository under accession code https://doi.org/10.5281/zenodo.7579177 [https://doi.org/10.5281/zenodo.7579177].

## Code availability
The model source code and the settings to reproduce the numerical simulations are available from the Zenodo repository under accession code https://doi.org/10.5281/zenodo.7579177 [https://doi.org/10.5281/zenodo.7579177] published under a GNU General Public License v3.0. Scripts used for data analysis and visualization are provided in the same repository we used Python (3.7.4) and the Python packages, scipy (1.3.1), pandas (1.1.4), numpy (1.19.4), matplotlib (3.1.1), and basemap (1.2.1). Spatial extrapolations of contaminated sites were performed using R (3.6.3) and the R libraries sf, maptools, raster, rgeos, geosphere, rgdal, spatstat, and ncdf4.

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

## Acknowledgements

We acknowledge the support of Michael Auer from the Heidelberg Institute for Geoinformation Technology (HeiGIT) for providing the OSMlanduse data. We thank Simone Stuenzi and Stephan Jacobi for supporting us with the graphical design of the figures. We acknowledge the support of Christina Himmelsbach for her support with the analysis of the contaminated site data. This work was supported by a grant from the German Federal Ministry of Education and Research (BMBF) awarded to M.L. (project PermaRisk, grant no. 01LN1709A). The work was also supported by the IceRoads project funded by the AWI Innovations Fonds (Innovation Project IP10200006). S.W. acknowledges funding through

Nunataryuk (EU Grant agreement no. 773421) and ESA Permafrost CCI (https://climate.esa.int/en/projects/permafrost/). R.Rolph was supported by the Geo.X, the Research Network for Geosciences in Berlin and Potsdam (Grant no. SO_087_GeoX). G.G. acknowledges support from EU Arctic Passion.

## Author contributions

M.L. designed the study, performed the data analysis, carried out the pan-Arctic simulations and analyses, prepared the figures and led the paper preparation. T.S.v.D. and G.G. co-designed the study and interpreted the results. S.W. helped with result interpretation and contributed to the development of the numerical models. R. Rolph provided background data on industrial contaminations in permafrost environments. R. Rutte performed the analysis of chemicals and environmental toxicities and provided important information on industrial infrastructure and waste management practices in the Arctic. M.S. provided interpretation and evaluation of land use data from O.S.M. V.R. provided input on the current environmental policy framework in the Arctic. S.A. performed the search in the Russian media and created the database of contamination incidents in Russia. A.O. supported the data analysis and data handling. All authors contributed to the writing and editing of the paper.

## Funding

## Competing interests

The authors declare no competing interests.
