## [Peer Review File · Nature Communications]

Thawing permafrost poses environmental threat to thousands of sites with legacy industrial contaminationReviewer #1 (Remarks to the Author):

Summary: Langer et al. analyze a database of sites across the Arctic domain that contain or store contaminants of various types. The authors evaluate the vulnerability of these sites to permafrost thaw to evaluate the potential releases of contaminants in the future. The authors locate the sites and categorize them by economic sector, the types of contaminants they contain, and toxicity. They then use projections of future conditions to identify those sites vulnerable to thaw. The authors conclude with a discussion of what we can do to monitor the sites. The study uses a unique combination of semantic analysis of metadata and physical modeling to isolate an issue rarely discussed in either the public arena or the scientific community. To me, this paper, and the few published papers like it, represent a starting point to determine how we respond to large-scale permafrost thaw.

I recommend accepting the paper with minor revisions. I find the data analysis and modeling techniques quite suitable for this analysis. My comments focus primarily on the interpretation and presentation of the results.

Specific Comments:

L42: The abstract needs to better identify the kinds of sites you discuss. All the sites contain contaminants of some sort, but not all are contaminated. Many represent storage facilities or working industrial facilities.

L88: Define exactly what you mean by 'access to industry data.' Do you want to fill in the uncertain category? Do you need the contaminants? The statement could be a political lightning rod because it implies industry has withheld data. Do they or has nobody asked for the data?

L93, Figure 1: I like the figure a lot, but it needs reworking by a graphics artist. The two panels contain too much blank space such that the individual site icons appear too small. The panels contain unessential elements that clutter the graphic, such as the ice wedges in the top panel and the blue arrows in the bottom panel. With some work, the authors could turn this figure into something everyone wants to download and use in a presentation.

L101: Rather than emphasizing the lack of information, the authors should emphasize they used all the available information. Emphasizing the lack of data undermines their case. This comment applies to the entire manuscript.

L113, Figure 2: The authors should use a simple pie chart for the percentages. Showing site density like in Figure 5a works much better than circle size. The larger circles make it look like the sites cover a huge area.

L137, Figure 3: Showing site density like in Figure 5a works much better than circle size.

L195: Define talik.

L224: The authors should state that we must extend the projections to reflect the time scale of permafrost. Those studying environmental contaminants have traditionally stopped their projections in 2050, but those studying biogeochemistry in permafrost regions extend the projections to 2200 or even 2300.

L228, Figure 5: The density map in Panel (a) is easier to understand than the bubble map. The authors should explain the mottled appearance in panel (b).

L264: I disagree. One can always make an assessment. The paucity of data makes the assessment more uncertain, not impossible.

L273: The authors should explain why this information does not exist. Is it because industry withholds it? Have governments even asked for the information? Does the information even exist?

L278: A global database is useful as long as it does not contain TBDs. The authors should identify exactly what information we need.

L281: Replace 'largely unpredictable' to 'highly uncertain.' One can always make a prediction. The lack of data simply makes it more uncertain. Besides, the authors make a prediction. The statement, as written, actually undermines their case.

L284: I suggest rephrasing to emphasize that better risk assessments will improve our ability to respond. I feel 'unprepared' is too strong a word here. A community with a waste dump and thawing permafrost already knows they have a big problem. They do not need a detailed inventory and risk assessment to tell them this.

L287: Delete Figure 6. The authors do not reference Figure 6 in the text and do not

explain it. This figure requires an entire paragraph to explain, but I do not feel it adds much value to the conclusions.

L314: The authors should emphasize that we need to develop new monitoring and mitigation strategies to deal with contaminants in permafrost. The sites will not leak randomly, as one might see in CONUS. Rather, an area undergoing large-scale thaw may have multiple sites leaking simultaneously. I agree we cannot install instrumentation to monitor every site, but what if we used remote sensing or we installed a single sensor as a surrogate to monitor multiple sites. Here is where the authors need to get the readers thinking outside the box.

L325: What do you mean by 'operational lifetime'?

L331: This single sentence likely highlights the most important conclusion of the paper, yet it does not appear in the abstract.

Reviewer #2 (Remarks to the Author):

This is a very interesting paper but will require some work to get it ready for publication. The paper opens with an Arctic focus but quickly moves into summarizing publicly available data from Alaska only. Was this because the data is the only available? I am still unsure. There are also many references to the methods so that in order to understand the paper I must bounce between sections. It would be better to make the paper easy to read, and a more linear story from start to finish. I made some specific comments below, but without additional clarity on the direction and focus of the paper, I won't be able to fully comment. I would encourage a rewrite, and then I would be delighted to review it again.

Line 53: Is this updated? The most recent number I saw was 5-8x faster

59: If you quantify in terms of 'latest' before the paper it dates your paper. Maybe quantify differently

72-77: excellent, I like the way you did this

78: Inducing? no sure what this sentence means

81: This is not the first paper to characterize waste locations by a long shot. Rephrase

88-92: excellent

101-103: Put (see methods) at the end of the sentence instead of twice in the body

103: Our database is a very strange way to characterize your study. Rephrase

105-110: If the IPCC scheme doesn't characterize the types of sites that you are interested in- why did you use it? This seems an arbitrary measurement to apply (unsuccessfully, as you state) that confuses the point of the paper

116: Is OSM a good source?

117-119:: same comment as above-- why use the IPCC scheme if it doesn't capture the nuance you are describing?

122: How does data from only one Arctic location solve the problem laid-out in 121?

128: citation needed

131: citation needed

134: citation needed

121-136: why summarize the data from the AK website in such detail/length? This is peculiar and doesn't add to the paper, but is distracting

Fig 3. citations needed

147: why is this striking? I would expect it

145-159: consider adding a table and significantly reducing this section. It is redundant of the earlier section that seemed out of place. I am starting to wonder if this paper is just about Alaska? If so, the intro needs to be re-written

Reviewer #3 (Remarks to the Author):

This study poses an interesting and significant pair of questions: (1) what is the number

of contaminated sites in permafrost regions in the northern circumpolar region; (2) when might these sites be affected by permafrost thaw associated with climate change? The study uses (often indirect) information on industrial sites around the Arctic, relationships between industrial sites and contamination from Alaska to derive transfer functions between industrial type and contamination, and RCP scenarios plus a ground temperature model to estimate when particular sites might be affected. These approaches appear logical and could be expected to yield a reasonable result. However, the study has what I view to be a fundamental weakness.

The only comprehensive data on the relationship between numbers of sites and those that are contaminated comes from Alaska. These data are used to extrapolate contamination to the entire Arctic. I cannot see how this approach is valid. Each of the nations included in the spatial extent of this study has its own history of industrial, resource and military development, its own regulatory regime (past and present), and its own enforcement regime for environmental regulations. For the study results to be valid, more than simply acknowledging this weakness (lines 182-185) is required. Without explicitly demonstrating that the Alaskan data is representative of the other nations – but particularly of Russia where the greatest amount of industrial development has occurred and which has large cities and more than half of the Arctic population - the results are speculative. The authors indicate that such data do not exist which suggests that adequate answers to the key questions are not possible at present for the entire Arctic, although they could be reasonably made for Alaska. The absence of any validation of the model results outside Alaska and the rather strange patterns of model results through the Canadian Arctic Archipelago (see below) do not create confidence.

Regrettably, therefore, I cannot recommend publication, nor can I see a way forward to revise the paper while its Pan-arctic focus is maintained.

Additional comments:

Line 101. I have no idea what "synergized" means in this sentence.

Figure 1. While some elements of this figure appear reasonable – drilling mud sumps, for example, others are not. Drilling rigs do not remain in place after a drill-hole has been completed and nor do excavation machines.

Figure 2 shows no sites in Greenland but Figure 5b shows high concentrations of contaminated sites.

Figure 4. Is the left to right positioning of the circles meaningful in this figure? I don't think it is, but this should be stated in the caption.

Figure 5. Potential errors in the spatial modelling are suggested by the strange patterns that exist across much of the region. For example, the Canadian Arctic Archipelago has only a handful of small communities, an even smaller number of military installations and some legacy exploration drill sites dating from the 1970s. I cannot reconcile this with the model output which yields variable density values of 10⁻³ to 10⁻⁴ km⁻² across extensive areas (Figure 5b).

REVIEWER COMMENTS

Reviewer #1 (Remarks to the Author):

Summary: Langer et al. analyze a database of sites across the Arctic domain that contain or store contaminants of various types. The authors evaluate the vulnerability of these sites to permafrost thaw to evaluate the potential releases of contaminants in the future. The authors locate the sites and categorize them by economic sector, the types of contaminants they contain, and toxicity. They then use projections of future conditions to identify those sites vulnerable to thaw. The authors conclude with a discussion of what we can do to monitor the sites. The study uses a unique combination of semantic analysis of metadata and physical modeling to isolate an issue rarely discussed in either the public arena or the scientific community. To me, this paper, and the few published papers like it, represent a starting point to determine how we respond to large-scale permafrost thaw. I recommend accepting the paper with minor revisions. I find the data analysis and modeling techniques quite suitable for this analysis. My comments focus primarily on the interpretation and presentation of the results. Specific Comments:

We like to thank the reviewer for the positive evaluation of our study. We are very thankful for all comments and suggestions. In the following we provide a point by point reply with all modifications made in the manuscript highlighted in bold.

L42: The abstract needs to better identify the kinds of sites you discuss. All the sites contain contaminants of some sort, but not all are contaminated. Many represent storage facilities or working industrial facilities.

We thank the reviewer for this suggestion and have improved the abstract accordingly:

We have identified about 4,500 industrial sites where potentially hazardous substances are actively handled or stored in the Arctic permafrost region. Furthermore, we estimate that between 10,000 and 20,000 contaminated sites are related to these industrial sites.

L88: Define exactly what you mean by 'access to industry data.' Do you want to fill in the uncertain category? Do you need the contaminants? The statement could be a political lightning rod because it implies industry has withheld data. Do they or has nobody asked for the data?

We thank the reviewer for this important comment. We fully agree that we should take the opportunity to clarify what information is already available and what data is missing. As part of the restructuring of the manuscript, we have now added an entirely new paragraph to the discussion that we believe is better suited to address this issue:

While our analysis provides an initial estimate of the magnitude of potentially substantial environmental risk, it also demonstrates the large uncertainty resulting from an incomplete database. We were able to reduce the data gap partly by using public media sources to compile a geospatial data subset on contaminated sites for the Russian Arctic. However, we note that there is still a great lack of public information on industrial sites and activities in the Arctic preventing a more comprehensive risk assessment. The data currently available are also insufficient to assess the likelihood of future contamination or the risk of its mobilization due to permafrost degradation. This would require more detailed information on the quantity and toxicity of industrially used substances and on the conditions of their storage or disposal. In particular, there is a lack of transparent documentation of environmentally hazardous substances transported to and used in the Arctic (31).

L93, Figure 1: I like the figure a lot, but it needs reworking by a graphics artist. The two panels contain too much blank space such that the individual site icons appear too small. The panels contain unessential elements that clutter the graphic, such as the ice wedges in the top panel and the blue arrows in the bottom panel. With some work, the authors could turn this figure into something everyone wants to download and use in a presentation.

We thank the reviewer for these recommendations. We have tried to address all the shortcomings of Figure 1 and hope that we managed to improve the Figure as suggested. If recommended by the editor we would hand this to a professional graphic designer:

Fig. 1: The potential impacts of thawing permafrost on above- and below-ground industrial infrastructure containing toxic substances or waste. The deepening of the thaw layer at the surface (active layer) unlocks frozen disposal sites and destabilizes foundations and containment structures. Furthermore, permafrost thaw intensifies thermo-hydrological erosion and increases the lateral flow of water, fostering the dispersion of contaminants.

L101: Rather than emphasizing the lack of information, the authors should emphasize they used all the available information. Emphasizing the lack of data undermines their case. This comment applies to the entire manuscript.

We agree with the reviewer that emphasizing the missing data is not as useful as describing what data were available and used and how we handled the resulting uncertainties. In the new version of the manuscript, we have rewritten and restructured the results section. The paragraphs presenting the datasets used now read:

We synthesized a geospatial dataset on industrial sites based on OpenStreetMap (OSM) and the Atlas of Population, Society and Economy in the Arctic (APSEA) which reveals insights into the spatial pattern of industrial activities in the Arctic permafrost region. Our database shows that about 4,500 land use elements labeled ‘industrial’ (hereafter called ‘industrial sites’) are located in the Arctic on continuous or discontinuous permafrost ground (Fig. 2). Following the International Panel of Climate Change (IPCC) classification scheme for industrial sectors, we find that most industrial sites (~65%) are either not labeled or cannot be assigned to one of the four major IPCC sectors ("Agriculture, Forestry and Other Land Use"; "Energy"; "Industrial Processes and Product Use"; "Waste") and are, thus, considered as “Uncertain”. Despite the incomplete labeling, the data provide a comprehensive and consistent assessment of the spatial density of industrial sites for the entire permafrost region of the Arctic.

In order to gain a better understanding of the relationship between industrial sites and the occurrence of contamination, we use regional data available for Alaska (Contaminated Sites Program, CSP (23)) and Canada (Federal Contaminated Sites Inventory, FCSI (24)). The synthesized dataset (CSP/FCSI) allows us to quantify the extent and nature of contamination and its spatial relationship to industrial sites (Fig. 3) for the permafrost region on the North American continent. We then extrapolate the spatial relationship between industrial sites and contaminated sites in the North American dataset to the panarctic, and validate the panarctic dataset with a compilation of Russian sites based on publicly available sources (see below).

L113, Figure 2: The authors should use a simple pie chart for the percentages. Showing site density like in Figure 5a works much better than circle size. The larger circles make it look like the sites cover a huge area.

We agree with the reviewer and have reworked the graphic. We have decided to illustrate the individual industrial site by simple points which give a good impression on the distribution of sites. The density maps are later used in the context of illustrating the spatial connection to the contaminated sites.

Fig. 2: The occurrence of industrial sites in the Arctic permafrost region (22). The database of industrial sites is based on OpenStreetMap (OSM) and the 2019 Nordregio Atlas of Population, Society and Economy in the Arctic (APSEA). While the industrial sectors "Energy" and "Agriculture, Forestry and Other Land Use" account for the largest proportion of industrial sites among the clearly labeled data, more than 60% of the mapped industrial sites are not clearly labeled. This creates a large uncertainty in quantifying specific industrial sectors while highlighting the need for improved databases on industrial activities in the Arctic.

L137, Figure 3: Showing site density like in Figure 5a works much better than circle size.

We agree with the reviewer that the circle sizes are difficult to read. However, we would like to present the direct point clustering when we have both datasets (industrial areas and contaminated sites) available. This way, we can reserve the density maps for presenting the results of the spatial modeling performed. We hope that the new maps will better illustrate the regional spatial distribution of industrial and contaminated sites.

Fig. 3.: The distribution of industrial and contaminated sites located on permafrost in (a) Alaska as reported by the Contaminated Sites Program (CSP) (23) and (b) Canada as reported by the Federal Contaminated Sites Inventory (FCSI) (24). Our classification "Cleanup complete" and "Not cleaned up" refers to information on whether hazardous substances have been removed or remained (or are suspected to remain) in the environment.

L195: Define talik.

Done:

In our model, permafrost is considered to degrade when simulations indicate the formation of a talik (defined here as a year-round unfrozen ground layer greater than 0.1 m thick).

L224: The authors should state that we must extend the projections to reflect the time scale of permafrost. Those studying environmental contaminants have traditionally stopped their projections in 2050, but those studying biogeochemistry in permafrost regions extend the projections to 2200 or even 2300.

We thank the reviewer for this reference. In the course of restructuring and reformulating the discussion, we have removed this statement at this point in favor of a modified later statement that reads as follows:

Our simulations underscore the need to extend the time horizon of previous risk studies, which have so far focused on mid-century (7). Risk studies need to consider the time scale (of centuries) of permafrost dynamics. The long-term consequences of past and present permafrost degradation will extend well into the second half of the 21st century and beyond, even if climate warming is halted.

L228, Figure 5: The density map in Panel (a) is easier to understand than the bubble map. The authors should explain the mottled appearance in panel (b).

Please note that we have rearranged the figures and also have added the following explanation to the caption:
The speckled appearance results from the clustering of industrial sites, especially in Russia and Canada.

L264: I disagree. One can always make an assessment. The paucity of data makes the assessment more uncertain, not impossible.

We agree with the reviewer and deleted the concerning statement in the revised discussion:

L273: The authors should explain why this information does not exist. Is it because industry withholds it? Have governments even asked for the information? Does the information even exist?

While we are not in a position to speculate on why data is not available for Russia, for example, we can use this study to demonstrate the value of openly available environmental data as in the case of Alaska and Canada. However, we agree with the reviewer that our statement requires further clarification of this issue, which we cover with the following statements:

The Contaminated Sites Program (CSP) in Alaska and the Federal Contaminated Sites Inventory (FCSI) in Canada demonstrate the value of transparent documentation of contamination status for science-based assessments. Without this comprehensive documentation, covering approximately 40% of the Arctic permafrost region, our large-scale assessment of contaminated sites in the Arctic permafrost region would not be possible.

In particular, there is a lack of transparent documentation of environmentally hazardous substances transported to and used in the Arctic (31). Moreover, there are no international environmental regulations for the Arctic, as formulated for the Antarctic in the Madrid Protocol, that require transparent documentation of contamination and potential sources of hazardous substances (32). At the same time, the lack of information is exacerbated by a rapidly growing number of industrial sites due to increasing economic interest and development in the Arctic (33).

L278: A global database is useful as long as it does not contain TBDs. The authors should identify exactly what information we need.

Again, we strongly agree with the reviewer. We have specified the required information of such a database and moved this statement to a later section:

The data currently available are also insufficient to assess the likelihood of future contamination or the risk of its mobilization due to permafrost degradation. This would require more detailed information on

the quantity and toxicity of industrially used substances, on its leachability potential (water solubility), proximity to hydrological pathways, the conditions of their storage or disposal, and the site-specific environmental conditions such as permafrost and ground stratigraphy. In particular, there is a lack of transparent documentation of environmentally hazardous substances transported to and used in the Arctic (31).

L281: Replace ‘largely unpredictable’ to ‘highly uncertain.’ One can always make a prediction. The lack of data simply makes it more uncertain. Besides, the authors make a prediction. The statement, as written, actually undermines their case.

We thank the reviewer for this comment. The paragraph in question has been completely rewritten. We have thoroughly reviewed our wording regarding uncertainties in the data and model results throughout the manuscript.

L284: I suggest rephrasing to emphasize that better risk assessments will improve our ability to respond. I feel ‘unprepared’ is too strong a word here. A community with a waste dump and thawing permafrost already knows they have a big problem. They do not need a detailed inventory and risk assessment to tell them this.

We agree with the reviewer and have decided to remove the statement in question in its entirety. This was also done to keep the discussion focused, as addressed in the reviewer's comment below.

L287: Delete Figure 6. The authors do not reference Figure 6 in the text and do not explain it. This figure requires an entire paragraph to explain, but I do not feel it adds much value to the conclusions.

Done. The Figure and the concerning paragraph was deleted (also see previous respond)

L314: The authors should emphasize that we need to develop new monitoring and mitigation strategies to deal with contaminants in permafrost. The sites will not leak randomly, as one might see in CONUS. Rather, an area undergoing large-scale thaw may have multiple sites leaking simultaneously. I agree we cannot install instrumentation to monitor every site, but what if we used remote sensing or we installed a single sensor as a surrogate to monitor multiple sites. Here is where the authors need to get the readers thinking outside the box.

We thank the reviewer for this suggestion. We have included this point by rewriting the entire paragraph:

In this context, it is also important to emphasize that permafrost thaw usually affects larger regions where there is a risk of multiple contaminated sites leaking at the same time. Therefore, technologies such as remote sensing, modeling, or selective monitoring of sites representative of larger regions could provide early warning for legacy sites. Consequently, we suggest that for contaminated industrial sites on permafrost, it is particularly important to reconsider the time, effort, and financial resources required for official long-term monitoring and site remediation. This includes the development of post-thaw scenarios. For most existing sites in the Arctic, the time period to consider may extend well into the second half of the 21st century and beyond due to lingering permafrost thaw. Our initial assessment suggests that existing and old industrial sites in the Arctic will be increasingly affected by climate change. The effects of thawing permafrost, with all its consequences such as loss of hydrological barriers, improved hydrological connectivity, reduced soil stability and strongly impeded site accessibility for clean-up measures, will often occur after the operating period of industrial sites. This underscores the need to avoid leaving environmentally hazardous substances at the sites, as permafrost can no longer be considered a reliable barrier to their containment. Furthermore, long-term remediation strategies will be necessary for contaminated legacy sites that have already been closed if they still contain hazardous substances.

L325: What do you mean by ‘operational lifetime?’

*The wording has been changed to **operating period** (see above)*

L331: This single sentence likely highlights the most important conclusion of the paper, yet it does not appear in the abstract.

We thank the reviewer for this remark and have added our conclusive statement to the abstract as:

Our analysis points to the serious environmental threat posed by the legacy of past and ongoing industrial

activities in the Arctic, which is exacerbated by the thawing of the permafrost. To avoid future environmental hazards, reliable long-term planning strategies for industrial and contaminated sites are needed that take into account the impacts of climate change.

Reviewer #2 (Remarks to the Author):

This is a very interesting paper but will require some work to get it ready for publication. The paper opens with an Arctic focus but quickly moves into summarizing publicly available data from Alaska only. Was this because the data is the only available? I am still unsure. There are also many references to the methods so that in order to understand the paper I must bounce between sections. It would be better to make the paper easy to read, and a more linear story from start to finish. I made some specific comments below, but without additional clarity on the direction and focus of the paper, I won't be able to fully comment. I would encourage a rewrite, and then I would be delighted to review it again.

We thank the reviewer for this constructive comment on the structure and flow of our manuscript. We would like to note that our manuscript has been significantly modified in response to Reviewer 3's comment encouraging us to further expand the contaminated sites database. We have located additional data on contaminated sites in Canada, so we can now base our estimates on a much larger database. Furthermore, we have collected data on contaminated sites in Russia in order to validate our extrapolation outside the North American continent. We also hope that in the revised version of the manuscript we have been able to clearly articulate the objectives of the study, which include both regional and global aspects of industrial sites and contaminated sites on permafrost soils in the Arctic. The objective of the study is to provide an initial assessment of the extent of industrial contamination in Arctic permafrost regions and to scope out the wider environmental risks associated with permafrost thaw.

Line 53: Is this updated? The most recent number I saw was 5-8x faster

We thank the reviewer for pointing this out. Indeed, there are different warming rates related to different temperatures and regions in the Arctic. We see that it is important to clarify this statement, so we now refer specifically to near-surface air temperature in the Arctic permafrost region, citing the most recent publication in this context.

In the Arctic permafrost region, near-surface air temperatures are rising at rates at least two times faster than the rest of the globe (1,2), with latest data analyses suggesting up to four-fold faster warming (3), substantially changing the ground stability and hydrological conditions (4,5).

59: If you quantify in terms of 'latest' before the paper it dates your paper. Maybe quantify differently

We agree. Thank you for raising the point, we have modified the sentence to:

One prominent environmental disaster attributed in part to loss of soil stability (10) was the huge diesel spill near the industrial city of Norilsk in northern Siberia in May 2020, where about 20,000 tons of fuel spilled from a destabilized tank facility and entered the Arctic ecosystem, thereby contaminating rivers, lakes, and tundra in a large permafrost watershed.

72-77: excellent, I like the way you did this

Thank you

78: Inducing? no sure what this sentence means

We wanted to describe the historical practice of depositing toxic substances into the soil. Perhaps we used the wrong term. Changed to:

Several experiments have been conducted in Alaska, Canada, and Russia in which toxic liquids and solids, including radioactive waste, have been deliberately deposited into permafrost for containment (17, 18, 19).

81: This is not the first paper to characterize waste locations by a long shot. Rephrase

Done:

Here we present a pan-Arctic compilation of the number of industrial sites on permafrost and derive a first panarctic estimate of the number of contaminated sites associated with these industrial sites. We also assess the different types of toxic substances associated with these activities in these regions. We further

use different future climate scenarios to evaluate how many and when these sites will be affected by permafrost thaw. We show that industrial legacy sites on thawing permafrost and the mobilization of toxic substances likely pose a significant environmental risk on the panarctic scale for which management strategies are needed.

88-92: excellent

Thank you

101-103: Put (see methods) at the end of the sentence instead of twice in the body

Corrected

103: Our database is a very strange way to characterize your study. Rephrase

*Changed to: **Our study***

105-110: If the IPCC scheme doesn't characterize the types of sites that you are interested in- why did you use it? This seems an arbitrary measurement to apply (unsuccessfully, as you state) that confuses the point of the paper

We thank the reviewer for pointing out that our description of how and why we use the IPCC label is misleading. The IPCC classes are used here to quantify the specific industry sectors. However, the information (completeness of labels) in the OSM database does not allow a clear classification for about 65% of the sites. This leads to a large uncertainty in the assignment to industrial sectors, but this is not a quality assessment of the database on the spatial distribution of industrial sites. Despite the lack of labels, the database provides comprehensive information on the spatial density of industrial sites on a global pan-Arctic scale. To clarify our intent, we have reworded and restructured the entire paragraph:

Following the International Panel of Climate Change (IPCC) classification scheme for industrial sectors ("Agriculture, Forestry and Other Land Use"; "Energy"; "Industrial Processes and Product Use"; "Waste"), we find that among the clearly labeled data, the classes 'Energy' and 'Agriculture, Forestry and Other Land Use' are the most dominant (both each >10%) in permafrost regions. However, we find that within the OSM-APSEA database most industrial sites (~65%) are either not clearly labeled or cannot be assigned to one of the four major IPCC classes and are, thus, considered as "Uncertain". While incomplete labeling leads to large uncertainties in quantifying specific industrial sectors, the dataset nevertheless provides a comprehensive and consistent assessment of the spatial distribution of industrial sites for the entire Arctic permafrost region. We are relating this spatial distribution of industrial sites to the occurrence of contaminated sites in the following assessment.

116: Is OSM a good source?

We are not aware of any other database that would provide consistent information on the location of industrial sites covering the entire Arctic. Since the OSM is in constant evolution and largely based on volunteered geographic information it is challenging to quantify its accuracy. In terms of completeness and accuracy the quality of OSM was recently evaluated for a limited set of land use features (Zhou et al, 2021). Unfortunately their analysis excluded industrial features. However, the general outcome of this quality assessment indicates that the completeness strongly varies among countries whereas the spatial accuracy is generally high. We are aware of the potential incompleteness of the applied dataset and describe these potential uncertainties in the Method section. In consequence we argue the available database provides a conservative estimate of the occurrence of industrial sites.

References:

Qi Zhou, Shuzhu Wang, Yaoming Liu, Exploring the accuracy and completeness patterns of global land-cover/land-use data in OpenStreetMap, Applied Geography (2022), doi: <https://doi.org/10.1016/j.apgeog.2022.102742>

The resulting database enabled a first-order assessment of the spatial distribution and concentration of industrial sites. However, historic industrial sites such as fully or partially revegetated old drill pads and

drilling sumps, covered landfills, and features yet undetected by remote sensing such as individual borehole sites, might be absent within our database. Furthermore, our database does not account for pipeline networks transporting oil and gas which span thousands of kilometers across the Arctic. Considering the omission of aforementioned objects within the OSM-APSEA database, we have produced a conservative estimate of industrial sites.

117-119:: same comment as above-- why use the IPCC scheme if it doesn't capture the nuance you are describing?

We have changed the figure caption to clarify our possibly misleading description of the IPCC class analysis:

Fig. 2: The occurrence of industrial sites in the Arctic permafrost region (22). The database of industrial sites is based on OpenStreetMap (OSM) and the 2019 Nordregio Atlas of Population, Society and Economy in the Arctic (APSEA). While the industrial sectors "Energy" and "Agriculture, Forestry and Other Land Use" account for the largest proportion of industrial sites among the clearly labeled data, more than 60% of the mapped industrial sites are not clearly labeled. This creates a large uncertainty in quantifying specific industrial sectors while highlighting the need for improved databases on industrial activities in the Arctic.

122: How does data from only one Arctic location solve the problem laid-out in 121?

In the revised version of our manuscript we explain in more detail how regional data on industrial and contaminated sites are used to generate a panarctic estimate (see above). Please also note that we have significantly expanded our data analysis by integrating additional data from Canada and validation data from Russia. We believe that our approach to deriving a pan-Arctic estimate of the number of contaminated sites on permafrost soils based on the spatial relationship between industrial and contaminated sites is now based on more robust data.

128: citation needed

Appropriate citation added

131: citation needed

Appropriate citation added

134: citation needed

Appropriate citation added

121-136: why summarize the data from the AK website in such detail/length? This is peculiar and doesn't add to the paper, but is distracting

We think that it is important to summarize the known facts that can be derived from the available data. Nevertheless, we agree with the reviewer that this should not distract from the central message of the study. Thus, we distilled this paragraph to the most relevant information:

The CSP/FCSI dataset shows about 8,000 individual contaminated sites (until January 2021) for Alaska and about 22,000 contaminated sites for Canada. About 12% (~3,600) of these sites are located in areas of continuous or discontinuous permafrost. Approximately 30% of the recorded contaminated sites in the permafrost region of the North American continent must be considered active. While the CSP and FCSI data provide relatively consistent information on site and treatment status, the two databases differ greatly in terms of information on industrial origin, historical record, and financial costs associated with site management. Therefore, we combine the two datasets only for the later analysis of the spatial correlation between industrial sites and the occurrence of contaminated sites. A detailed analysis on toxic substances and their industrial origin focuses on Alaska as an example.

Fig 3. citations needed

Appropriate citation added

147: Why is this striking? I would expect it

We have removed the word "striking" to focus on the pure results.

145-159: consider adding a table and significantly reducing this section. It is redundant of the earlier section that seemed out of place. I am starting to wonder if this paper is just about Alaska? If so, the intro needs to be re-written

We agree with the reviewer that the structure of the original manuscript was not as condensed and focused as it needed to be to convey the main message of our study. We completely restructured and reworded the "Results" and "Discussion" sections. In addition, the Executive Summary and Introduction were changed. This was also necessary because the new version of the manuscript now takes into account a whole new database with data from Canada and Russia. We very much hope that the new manuscript now more clearly presents the scope of the study and its conclusions.

Reviewer #3 (Remarks to the Author):

This study poses an interesting and significant pair of questions: (1) what is the number of contaminated sites in permafrost regions in the northern circumpolar region; (2) when might these sites be affected by permafrost thaw associated with climate change? The study uses (often indirect) information on industrial sites around the Arctic, relationships between industrial sites and contamination from Alaska to derive transfer functions between industrial type and contamination, and RCP scenarios plus a ground temperature model to estimate when particular sites might be affected. These approaches appear logical and could be expected to yield a reasonable result. However, the study has what I view to be a fundamental weakness.

We thank the reviewer for the critical analysis of our study. We emphasize that we are aware of the uncertainties in our study, which are openly presented. Nevertheless, we would like to point out that we use as much direct data as is currently available. In particular, the data on contaminated sites are based on quality-controlled and continuously updated sources. However, we thank the reviewer for his thorough review, which has stimulated us to conduct a new compilation of contaminated site data that now includes Alaska and Canada. In addition, we now also collected data from Russia for validation purposes. We hope that with this new substantially expanded dataset and the analysis performed, we can convince the reviewer that our study provides a solid first estimate of the number and the spatial distribution of contaminated sites in the Arctic permafrost region.

The only comprehensive data on the relationship between numbers of sites and those that are contaminated comes from Alaska. These data are used to extrapolate contamination to the entire Arctic. I cannot see how this approach is valid. Each of the nations included in the spatial extent of this study has its own history of industrial, resource and military development, its own regulatory regime (past and present), and its own enforcement regime for environmental regulations. For the study results to be valid, more than simply acknowledging this weakness (lines 182-185) is required.

We agree with the reviewer that our previous analysis was based on a relatively small regional dataset, and also agree that regional/national differences should be accounted for by using a more broadened database. Therefore, in addition to the Alaska dataset already used and analyzed, we have now included contaminated site data from the Canadian Federal Contaminated Sites Inventory (FCSI). The new larger dataset results in much more nuanced projections between point process models, which are now fitted to data with regional differences across the North American continent. The spatial relationship found previously remains essentially the same. However, the new extrapolations now show a slightly increased number of contaminated sites (especially for Russia). At the same time, the spread between the models has increased, as the new database now accounts for regional differences across the entire permafrost region of the North American continent.

Without explicitly demonstrating that the Alaskan data is representative of the other nations – but particularly of Russia where the greatest amount of industrial development has occurred and which has large cities and more than half of the Arctic population – the results are speculative. The authors indicate that such data do not exist which suggests that adequate answers to the key questions are not possible at present for the entire Arctic, although they could be reasonably made for Alaska.

In addition to the comprehensive data from Alaska and Canada, we now have created a database for contaminated sites in Russia based on publicly available data and media sources. Although our Russian database is certainly not complete, it provides a subsample that can be used to validate the spatial relationships between industrial and contaminated sites predicted by the models. By randomly subsampling the model results, we generate artificial observations that can be compared and “ground-truthed” with the real observations. We find that the model reproduces well the observed distribution. The observed number of contaminated sites is always within the plausible range of the predicted number of contaminated sites. This suggests that a similar spatial relationship between industrial sites and contaminated sites as in North America can be expected for Russia. The results of this validation are displayed in the supplement:

Figure S7: The bars show the number of surveyed contaminated sites in Russia that are located within the predicted intensity classes of the two point process models. To verify that the observed number of contaminated sites matches the expected number according to the models, multiple ($N=1000$) random samples ($n=44$) are drawn from the entire modeled distribution for Russia (shown as a boxplot). The models are found to reproduce the observed distribution well, mostly within the interquartile ranges (boxes) and always within 1.5 times the interquartile ranges (whiskers).

Further, we have added the following section to the Methods:

To validate the applicability of the point process models outside the North American continent, we create a sample dataset on industrial contamination in the permafrost region of Russia. The dataset was created based on a Google search using keywords in Russian ("загрязнение" - pollution; "разлив нефтепродуктов" - oil spill; "техногенная авария" - industrial accident or disaster; "Арктика" - Arctic; "мерзлота" - permafrost) and a search of several online media (including local and federal news portals). The database includes the locations of contamination events, the date, and information about the type of event, as well as the link to the media source. In some cases, the contaminated area and volume of the spill are also provided. However, we point out that it is mostly not possible to independently verify these numbers. In order to avoid a bias due to expectation, the data search was performed without considering the map of projected contaminated sites. In total we obtained 102 events, starting mostly from the 2000s. As the search included either a general term "permafrost" or "Arctic", only 44 events fell into the zones of continuous and discontinuous permafrost, leaving another 58 events in the zones of sporadic permafrost, isolated patches or no permafrost. We analyze their location in relation to the modeled intensity maps for Russia (Fig. 5c). To this end, we apply logarithmic binning to the model results for Russia to spatially summarize the model results into six intensity classes. For each intensity class, we calculate the expected total number of contaminated sites by spatial integration. We draw ($N=1000$) random samples from the modeled total distribution, each comprising the number of observed contaminated sites ($n=44$). The resulting distributions for each intensity class are then compared to the observed contaminated sites and their occurrence within modeled intensity class (SI Appendix, Fig. S7). The model-based random selection reproduces the observed distribution very well. The observed number of contaminated sites across all intensity classes is always within the plausible range of the predicted

number of contaminated sites. This suggests that the spatial relationship between industrial sites and contaminated sites found for the permafrost region of the North American continent also applies to the Russian permafrost region.

The absence of any validation of the model results outside Alaska and the rather strange patterns of model results through the Canadian Arctic Archipelago (see below) do not create confidence.

We hope that the extensively expanded database, together with the validation performed for Russia, will provide more confidence in our analyses and projections. The pattern in the Canadian Arctic Archipelago results from the spatial occurrence of industrial sites in this region. The new data confirm that the spatial relationship between industrial and contaminated sites also applies to Canada with an uncertainty range of about 16% for the model results. The map of contaminated sites for Canada also confirms this pattern.

Regrettably, therefore, I cannot recommend publication, nor can I see a way forward to revise the paper while its Pan-arctic focus is maintained.

We have significantly expanded the database for our analysis and can now provide a very robust estimate of contaminated sites in the permafrost regions of the Arctic. Our database for the performed extrapolation now covers the entire North American continent, and for Russia we have compiled a validation dataset from multiple available sources. Using these data, we can show that the relationship between industrial sites and contaminated sites in the three largest Arctic countries follows very similar, predictable spatial patterns. Although our estimate of the total number of contaminated sites among the industrial sites is still subject to large uncertainties, we are able to provide novel insights into the extent of industrial contamination for the entire Arctic permafrost region.

Additional comments:

Line 101. I have no idea what “synergized” means in this sentence.

We apologize for the wrong wording and rephrased the sentence to:

We synthesized a geospatial dataset on industrial sites based on OpenStreetMap (OSM) and the Atlas of Population, Society and Economy in the Arctic (APSEA) which reveals insights into the spatial pattern of industrial activities in the Arctic permafrost region.

Figure 1. While some elements of this figure appear reasonable – drilling mud sumps, for example, others are not. Drilling rigs do not remain in place after a drill-hole has been completed and nor do excavation machines.

We thank the reviewer for pointing to the unrealistic elements in our schematic figure. We have modified the figure also in agreement with comments from reviewer 1.

Fig. 1: The potential impacts of thawing permafrost on above- and below-ground industrial infrastructure containing toxic substances or waste. The deepening of the thaw layer at the surface (active layer) unlocks frozen disposal sites and destabilizes foundations and containment structures.

Furthermore, permafrost thaw intensifies thermo-hydrological erosion and increases the lateral flow of water, fostering the dispersion of contaminants.

Figure 2 shows no sites in Greenland but Figure 5b shows high concentrations of contaminated sites.

We thank the reviewer for pointing us to this error. There was an obvious failure during generating the figure from the GIS. Figure 2 has been reworked and corrected also because of the comment from reviewer 1 asking for a different way of illustrating the data.

Fig. 2: The occurrence of industrial sites in the Arctic permafrost region (22). The database of industrial sites is based on OpenStreetMap (OSM) and the 2019 Nordregio Atlas of Population, Society and Economy in the Arctic (APSEA). While the industrial sectors "Energy" and "Agriculture, Forestry and Other Land Use" account for the largest proportion of industrial sites among the clearly labeled data, more than 60% of the mapped industrial sites are not clearly labeled. This creates a large uncertainty in quantifying specific industrial sectors while highlighting the need for improved databases on industrial activities in the Arctic.

Figure 4. Is the left to right positioning of the circles meaningful in this figure? I don't think it is, but this should be stated in the caption.

We thank the reviewer for this comment and have added an explanation to the caption, which now reads as follows:

Figure 4: Most frequently occurring toxic substances at contaminated sites in permafrost in Alaska as reported by the Contaminated Sites Program (CSP). The occurrence is further differentiated according to the industrial sectors Industrial Processes and Product Use (IPPU), Energy, Military, and Waste. Toxic substances from Agriculture, Forestry, and Other Land Use (AFOLU) occur in negligible numbers. Substances are ordered vertically according to their toxicity, which was assessed using the median lethal concentration for fish (LC50-fish) after 96 hours (see also SI Appendix, Fig. S3 and Tab. S1). Please note that the lateral position is arbitrary.

Figure 5. Potential errors in the spatial modelling are suggested by the strange patterns that exist across much of the region. For example, the Canadian Arctic Archipelago has only a handful of small communities, an even smaller number of military installations and some legacy exploration drill sites dating from the 1970s. I cannot reconcile this with the model output which yields variable density values of 10^{-3} to 10^{-4} km⁻² across extensive areas (Figure 5b).

We thank the reviewer for giving special consideration to Canadian Arctic Archipelago and thus questioning the model results. The new version of the manuscript now uses a new database on contaminated sites including Canada. The database confirms that also for the Canadian Arctic Archipelago a considerable number of contaminated sites is reported which follow the spatial pattern of industrial sites (see new figure below).

Figure 5: Density maps of contaminated sites per area as derived by the point process models for the permafrost regions of (a) Alaska and (b) Canada with dots depicting the locations of industrial and contaminated sites. Map (c) shows the predicted density of contaminated sites for the permafrost region in Russia. The dots show the data on contaminated sites surveyed in this study and that are used for model validation. Applying the model to the entire Arctic permafrost region yields a pan-Arctic map of estimated contaminated site density (d). Note that the speckled appearance results from the regional clustering of industrial sites combined with the chosen spatial resolution (50 km) of the filter used for the point process models.

Reviewer #2 (Remarks to the Author):

This is overall a good revision. I still think that using 'open street map' instead of one of the excellent satellite databases put out by NASA or ESA is a critical methodology mistake. Otherwise, good corrections.

Reviewer #3 (Remarks to the Author):

The manuscript is greatly improved from the first version with the addition of the Canadian data and more limited data from Russia. The extension of the conclusions about industrial sites and contaminated sites to the Arctic as a whole is now justifiable. However, I recommend some further revisions to deal with my comments below. A crucial element of the analysis is the definition of the spatial domain to which it applies. This is frequently but inconsistently referred to in the manuscript, as continuous and discontinuous permafrost, the permafrost region..., the panarctic region, and so on. I have highlighted most (but perhaps not all) of the many cases where this pertains in the ms.. What I understand from the methods and Figure 2 is that the data and models apply to areas where the probability of permafrost is greater than 50%. In traditional permafrost parlance this would mean the discontinuous permafrost zone (50-90% of the area underlain by permafrost) and the continuous permafrost zone (>90% underlain by permafrost). It's important to note that this does not mean discontinuous permafrost and continuous permafrost as sometimes stated in the manuscript, as these are concepts rather than geographical areas. In particular, permafrost is discontinuous right to its southernmost extent where it occupies only 1% of the landscape. The permafrost region encompasses all the permafrost zones, including the zones of sporadic discontinuous permafrost and isolated patches, so if I've understood the methods correctly, permafrost region should not be used. Instead, I would recommend using "permafrost model domain" and then define this early and clearly in the paper as being the areas where permafrost probability is estimated to be >0.5. However, other terminological choices could be made.

A second critical element that needs clarification is the modelling to predict current and future spatial and temporal trends in thermal state. It is unclear (at least to me) how the permafrost temperature modelling relates to the spatial model of permafrost distribution which was used to establish the spatial domain. There is a significant difference in scale between the two approaches, with the past and future modelling at 1° resolution (about 100 x 100 km) and the permafrost probability model (Obu et al.) having been run at a 1 km² resolution. I believe, but I am not sure because this section of the methods is quite brief, that the CryoGridLite was run just for the domain which had been demarcated by Obu et al. as having a permafrost probability of 0.5 or greater. But there is no comparison of the output of the two.

If I have correctly understood the methods and assumptions regarding permafrost thaw, the following seems to be argued:

- 1. Permafrost in the discontinuous permafrost zone is degrading under the current climate (a statement that may be true in a gridded model but is not universally correct in the field).**
- 2. Degradation is identified in the simulations by the formation of a supra-permafrost talik.**
- 3. Therefore every modelled location which has a supra-permafrost talik in the simulations is part of the discontinuous permafrost zone.**
- 4. Industrial and contaminated sites which fall into the modelled zone of discontinuous permafrost are therefore already being subjected to thaw.**
- 5. Changes in the distribution of modelled grid cells with and without taliks between 1960 and 2000 affected 40% of the industrial sites and 20% of contaminated sites indicating that many sites were impacted over this period.**

If the parameter settings used in CryoGridLite did not produce ground temperatures that align well spatially with the permafrost area shown in Obu et al., then this would have a significant impact on the projected timing and spatial extent of permafrost degradation. In my opinion, a comparison is required to provide confidence that the projected

permafrost conditions and the formation of taliks are reasonable. While it is true that the authors describe their modeling "a first-order assessment", this part of the research is fundamental to the conclusions of the study regarding when and where sites will be impacted.

The paper is generally well-written, but I have edited the text with the changes tracked to help smooth the English. The authors can adopt these suggestions or not as they wish.

Figure 4 remains problematic in my view. It seems to show similar information to Figure S3. I appreciate the addition of the final sentence in the caption in response to my initial review but I still find the positioning of the circles from left to right quite distracting. I would recommend replacing this figure with S3 which is immediately understandable and actually includes more information.

I have also added some additional generally minor comments on the manuscript file.

Antoni Lewkowicz

In their review of the second version of this manuscript, reviewer #3 added some comments to the manuscript file and supplementary information file. These comments were forwarded to the authors, who replied as included in this Peer Review File.

Reviewer #2 (Remarks to the Author):

This is overall a good revision. I still think that using 'open street map' instead of one of the excellent satellite databases put out by NASA or ESA is a critical methodology mistake. Otherwise, good corrections.

We would like to thank the reviewer for the positive assessment of our revisions. We would like to take this opportunity to explain why, despite known shortcomings and uncertainties in OSM data, we and other recent studies use it to estimate infrastructure occurrence and distribution. To date, OSM is the only database available that allows us to assess the location of industrial sites worldwide. There are more advanced products based on satellite data such as the SATCHI database, but this product is currently limited to the Arctic coastal zone (up to 100 km inland). Since a large proportion of industrial sites and contaminated sites are located further inland, our analysis requires a larger coverage. This is also why many other recent studies based on infrastructure data in the Arctic still entirely rely on OSM data (e.g. Hjort et al., 2018, Suter et al., 2019, Ran et al., 2022, Liew et al., 2022). However, we agree with the reviewer that we can use the more advanced products such as SATCHI to assess the uncertainty within our OSM-APSEA database. The revised version of the manuscript now includes a comparison that highlights the weaknesses and strengths of our database. We find that the total number of industrial sites is most likely underestimated, but that the general spatial occurrence of industrial sites is well captured. We emphasize that our extrapolation of contaminated sites is based on relative differences in the spatial density of industrial sites. For generating maps of spatial density differences, the absolute number of sites is less important than their relative proportions. We find that despite remaining uncertainties, OSM provides a reasonable basis for our analysis. However, we would like to emphasize that more advanced products becoming available in the future will certainly allow a finer and more precise analysis of the spatial relationship between industrial and contaminated sites in the future, and we look forward to their publication.

In the method section:

The completeness of the combined OSM-APSEA database on industrial sites was assessed by comparison with the Arctic Coastal Human Impact (SACHI) dataset (Bartsch et al., 2021), which provides a map of infrastructure areas along Arctic shorelines impacted by permafrost (limited to 100 km inland). To compare the spatial consistency in terms of the occurrence of industrial sites between the two datasets, we allowed a buffer radius of 100 m around the OSM-APSEA point data. In this way, the existing positives from OSM-APSEA were evaluated for the entire SACHI domain. Missing positives were only evaluated for selected test regions (in Alaska, Canada, Greenland, and Russia), as it was necessary to manually select the infrastructure areas within SACHI that represent compatible industrial sites. The assessment showed that the presence of industrial sites in the OSM-APSEA database is more than 85% consistent with the presence of industrial infrastructure in the SACHI database for the test regions and the entire SACHI domain. However, OSM-APSEA has a much lower number of industrial sites compared to SACHI (73% missing). The observed discrepancies are consistent with values from previous comparisons between SACHI and OSM (Bartsch et al., 2021). Based on the database comparisons, we argue that OSM-APSEA provides a relatively reliable estimate of the regional presence of industrial sites in the Arctic, but that there is substantial uncertainty regarding the absolute number of individual infrastructure elements. Given this analysis and the fact that the above mentioned infrastructure elements were not included in the OSM-APSEA database, we assess our estimate of industrial sites as conservative.

Figure S6: A comparison between the OSM-APSEA database and the Arctic coastal infrastructure satellite product (SACHI) reveals a match of more than 85% (true-positive) for selected test regions as well as for the total SACHI domain (a). This means that wherever OSM-APSEA indicates an industrial site, the SACHI database indicates in more than 85% of the cases the presence of industrial infrastructure. For the test regions, individual industrial infrastructure elements were manually classified within the SACHI dataset to facilitate a direct comparison between the number of industrial sites and the number of infrastructure elements (b). This comparison shows that the number of industrial sites within OSM-APSEA is on average $40 \pm 20\%$ lower than the number of industrial infrastructure elements in SACHI. For the test regions the bias might be correct by a simple multiplier.

Reviewer #3 (Remarks to the Author):

The manuscript is greatly improved from the first version with the addition of the Canadian data and more limited data from Russia. The extension of the conclusions about industrial sites and contaminated sites to the Arctic as a whole is now justifiable.

We thank Antoni Lewkowicz for his valuable comments and suggestions, which led to a greatly improved study.

However, I recommend some further revisions to deal with my comments below. A crucial element of the analysis is the definition of the spatial domain to which it applies. This is frequently but inconsistently referred to in the manuscript, as continuous and discontinuous permafrost, the permafrost region..., the panarctic region, and so on. I have highlighted most (but perhaps not all) of the many cases where this pertains in the ms..

We agree with the reviewer that there were some inconsistencies in terminology for the study domain in our manuscript. We followed the reviewer's suggestions in the manuscript and thoroughly reviewed the entire text to remove inconsistent terminology of the studied spatial domain. We hope that the revised version of the manuscript is now using a clear terminology for the delineation of the spatial domain covered.

What I understand from the methods and Figure 2 is that the data and models apply to areas where the probability of permafrost is greater than 50%. In traditional permafrost parlance this would mean the discontinuous permafrost zone (50-90% of the area underlain by permafrost) and the continuous permafrost zone (>90% underlain by permafrost). It's important to note that this does not mean discontinuous permafrost and continuous permafrost as sometimes stated in the manuscript, as these are concepts rather than geographical areas. In particular, permafrost is discontinuous right to its southernmost extent where it occupies only 1% of the landscape. The permafrost region encompasses all the permafrost zones, including the zones of sporadic discontinuous permafrost and isolated patches, so if I've understood the methods correctly, permafrost region should not be used. Instead, I would recommend using "permafrost model domain" and then define this early and clearly in the paper as being the areas where permafrost probability is estimated to be >0.5. However, other terminological choices could be made.

We agree with the reviewer that the terminology used in our manuscript did not correctly use the terms continuous and discontinuous permafrost. In fact, we used the probability of permafrost occurrence as modeled by Obu et al. to delineate our study area. We focus on regions where the spatial probability of permafrost occurrence is greater than 50% and thus focus on regions that are dominated by permafrost. To avoid confusion, we now omit the terms continuous and discontinuous permafrost when referring to our simulations and analyses. We have adopted the reviewer's suggestion and use the term "permafrost model domain".

A second critical element that needs clarification is the modelling to predict current and future spatial and temporal trends in thermal state. It is unclear (at least to me) how the permafrost temperature modelling relates to the spatial model of permafrost distribution which was used to establish the spatial domain. There is a significant difference in scale between the two approaches, with the past and future modelling at 1° resolution (about 100 x 100 km) and the permafrost probability model (Obu et al.) having been run at a 1 km² resolution. I believe, but I am not sure because this section of the methods is quite brief, that the CryoGridLite was run just for the domain which had been demarcated by Obu et al. as having a permafrost probability of 0.5 or greater. But there is no comparison of the output of the two.

We thank the reviewer for pointing us to the unclear description of our modeling approach and its relationship to the spatial delineation of the permafrost region under study. As now more precisely explained in the manuscript, we used the permafrost probability model (Obu et al.) to delineate industrial and contaminated sites currently located in the permafrost-dominated region. For the transient modeling of permafrost, we used the CryoGridLite model, which was run at a coarse spatial resolution of 1 degree due to its high computational cost. The model was run on a domain defined by

the Northern Circumpolar Soil Carbon Database version 2 (Hugelius et al., 2013) which essentially determines the ground stratigraphies used by the model. While we agree that the model description is very short we like to keep it this way because the permafrost thaw projections are only one component in our study. However, a detailed technical description and validation of the model is provided in (Langer et al., 2022).

We agree with the reviewer that we have missed demonstrating the consistency between the permafrost probability map (which we have used to delineated our study region for the geospatial analysis and extrapolation), and the applied transient modeling to project the future evolution of permafrost at the industrial and contaminated sites. In the supplementary information, we now demonstrate how the simulated permafrost occurrence probability relates to simulated talik occurrence under current climate conditions.

Figure S8: The map (a) shows the outline of the permafrost model domain delineated by permafrost occurrence probabilities (>50%) using the Northern Hemisphere Permafrost Map (NHPM) (22), underlain by persistent talik presence as simulated with the CryoGridLite model for the reference period (2000-2016). The histogram (b) shows the distributions of permafrost occurrence probabilities from the NHPM (20) for the CryoGridLite grid cells stimulated as talik-free and those with talik or permafrost-free between 2000 and 2016. Permafrost occurrence probabilities from the NHPM (20) were aggregated to the spatial resolution of the CryoGridLite grid cells (1 degree) by averaging.

and added to the methods the following:

To evaluate the consistency between the NHPM (22) delineated permafrost model region and the CryoGridLite permafrost simulations, we compare the NHPM permafrost occurrence probabilities with the simulated presence or absence of persistent taliks (Fig. S8). Regions with taliks and permafrost free areas were combined for this purpose. The comparison was made for the same reference period (2000-2016) and the results show that the majority of grid cells (80%) with simulated talik occurrence are located in regions where the probability of permafrost occurrence is low (less than 50%), while the majority of grid cells (80%) with simulated talik-free conditions are located in regions where the probability of permafrost occurrence is high (more than 90%). These limits align well with the values used to distinguish zones of continuous and discontinuous permafrost within the NHPM.

Please note that in the revised version we also clarify the resolution difference between the CryoGridLite simulations (run on 1° degree resolution) and the Northern Circumpolar Soil Carbon Database version 2 (max resolution 1 km). We also clarify what this means for the interpretation of the simulations.

In the main text:

The model has coarse spatial resolution (1 degree), and thawing of permafrost within grid cells

is considered likely if simulations indicate the formation of a talik (defined here as a year-round unfrozen soil layer greater than 0.1 m thick above permafrost).

In the method section:

The modeling performed should be considered as a first-order assessment of the mean thermal state of the soil within the respective model grid cells, which do not represent permafrost conditions at specific locations.

If I have correctly understood the methods and assumptions regarding permafrost thaw, the following seems to be argued:

1. Permafrost in the discontinuous permafrost zone is degrading under the current climate (a statement that may be true in a gridded model but is not universally correct in the field).
2. Degradation is identified in the simulations by the formation of a supra-permafrost talik.
3. Therefore every modelled location which has a supra-permafrost talik in the simulations is part of the discontinuous permafrost zone.
4. Industrial and contaminated sites which fall into the modelled zone of discontinuous permafrost are therefore already being subjected to thaw.
5. Changes in the distribution of modelled grid cells with and without taliks between 1960 and 2000 affected 40% of the industrial sites and 20% of contaminated sites indicating that many sites were impacted over this period.

If the parameter settings used in CryoGridLite did not produce ground temperatures that align well spatially with the permafrost area shown in Obu et al., then this would have a significant impact on the projected timing and spatial extent of permafrost degradation. In my opinion, a comparison is required to provide confidence that the projected permafrost conditions and the formation of taliks are reasonable. While it is true that the authors describe their modeling “a first-order assessment”, this part of the research is fundamental to the conclusions of the study regarding when and where sites will be impacted.

We thank the reviewer for his thorough assessment of our chain of reasoning, which highlights the importance of accurate definitions for an analysis of current and future permafrost stability. We fully agree with the reviewer that it is critical that both the permafrost map and the transient modeling results are consistent (see also response above). For this reason, we have now made a comparison between the permafrost probability map and our own transient simulations for the same time period (2000-2016). The simulations agree well in terms of the regional extent of stable permafrost (probability of permafrost occurrence > 90% = talik free in the model). We acknowledge that it is complex and potentially ambiguous to extract subgrid permafrost distribution and talik formation from a single run per grid cell. We also fully agree with the reviewers' earlier comment that our model does not represent the heterogeneous conditions of degrading permafrost, but only provides a very general indication of regional thermal state, we have added the notation "in model" to the terms "talik" and "talik free" to make it clear that our interpretation is purely model-based.

In the light of the reviewer's evaluation we have reviewed and adapted our analysis strategy. Following the recommended consistency check we now changed the script from producing the first talik occurrence date to the date when talik occurrence became persistent, as this is found more consistent with the permafrost occurrence probabilities of Obu et al.

Accordingly, we re-analyzed the simulations and updated the method and results sections:

Figure 6: Number of industrial sites and number of upscaled contaminated sites located in zones of the permafrost model domain affected by permafrost thaw (a) based on RCP 2.6 and RCP 8.5 warming scenarios (model mean from the CMIP5 projections based on CCSM4 and HADGEM2-ES). The shaded areas show the uncertainty range due to the spatial extrapolations based on the two point process models (PPM1 and PPM2). The related global temperature increase is shown as an anomaly compared to the pre-industrial period (1850-1900) (b).

The effects of the updated analysis are:

- *The number of sites affected by permafrost thaw under past and current climate conditions (1960-2020) has decreased. All statements and results were updated accordingly.*
- *The total number of industrial and contaminated sites potentially affected by permafrost thaw between 2050 and 2100 remains largely unchanged.*
- *The uncertainty in simulated permafrost destabilization (persistent talik occurrence) has decreased.*

The reason is that the updated talik definition leads to a later date when permafrost is expected to become unstable. Since the data of persistent talik occurrence is less affected single years the uncertainty strongly decreases. The updated talik analysis is still based on the same permafrost simulations that project a substantial decrease in the permafrost dominated region as the climate warms, affecting almost all industrial and contaminated sites. This explains why, towards the end of the projection period, the total number of industrial and contaminated sites affected by permafrost thaw remains the same.

We emphasize that the main conclusions of the study remain unchanged and are:

- *There are at least 4,500 industrial sites in permafrost-affected regions of the Arctic where potentially hazardous materials are handled or stored.*
- *There are between 13,000 and 20,000 contaminated sites associated with these industrial sites.*
- *Permafrost will thaw at thousands of industrial sites and contaminated sites by the end of this century.*

The paper is generally well-written, but I have edited the text with the changes tracked to help smooth the English. The authors can adopt these suggestions or not as they wish.

We are very thankful for these edits and suggestions made by the reviewer which we have largely adopted.

Figure 4 remains problematic in my view. It seems to show similar information to Figure S3. I appreciate the addition of the final sentence in the caption in response to my initial review but I still find the positioning of the circles from left to right quite distracting. I would recommend replacing this figure with S3 which is immediately understandable and actually includes more information.

We agree with the reviewer and have replaced the figure as suggested:

Figure 4: Most frequently occurring toxic substances at contaminated sites in the permafrost dominated regions of Alaska as reported by the Contaminated Sites Program (CSP). The occurrence is further differentiated by industrial sector: Industrial Processes and Product Use (IPPU), Energy, Military, and Waste. Toxic substances from Agriculture, Forestry, and Other Land Use (AFOLU) occur in negligible numbers. The toxicity of each substance is indicated using the median lethal concentration for fish (LC₅₀-fish) after 96 hours (see also SI Appendix, Tab. S1).

I have also added some additional generally minor comments on the manuscript file.
Antoni Lewkowicz

Again we would like to thank Antoni Lewkowicz for his valuable, thorough, and supportive review. His comments and suggestions have greatly improved our study and we hope that concerns are adequately addressed. The specific comments and suggestions together with our modifications are listed in the following:

Specific comments of the reviewer made within the manuscript:

Main text

Do you know that these are all on permafrost or in the permafrost region?

Statement changes to:

Ongoing climate warming will significantly increase the risk of contamination and mobilization of toxic substances since about 1,100 industrial sites and 3,500 to 5,200 contaminated sites currently located in regions of stable permafrost will start to thaw before the end of this century.

This is not just the active layer. When the active layer does not refreeze, this is a talik. Therefore, better to omit the term active layer here. You do not discuss that permafrost at temperatures close to 0°C loses bearing capacity so destabilization of foundations can occur even while permafrost remains. Furthermore, at contaminated sites, porewater may be sufficiently solute-rich that substantial movement of groundwater may occur at temperatures below 0°C. Adding these ideas would further strengthen your arguments.

We have modified the caption accordingly:

Fig. 1: The potential impacts of thawing permafrost on above- and below-ground industrial infrastructure containing toxic substances or waste. The warming and thaw of near surface permafrost unlocks frozen disposal sites and destabilizes foundations and containment structures. Furthermore, permafrost thaw intensifies thermo-hydrological erosion and increases the lateral flow of water, fostering the dispersion of contaminants.

and we have added as suggested the impact on bearing capacity and the freezing-point depression due in solute-rich environments to the discussion.

We emphasize that permafrost already loses substantial bearing capacity at temperatures close to 0°C, so that destabilization of any infrastructure may occur even if permafrost is assumed to be stable according to our simulation. This aspect is further underlined when considering that permafrost degradation can be strongly enhanced through the impact of infrastructure on ground thermal conditions which is not accounted for in this study.

Should clarify the caption so that it's obvious that reference 22 relates to the permafrost map only. It's important to clearly state that the map does not show the full permafrost region (as specified in the caption) but only the area >50% probability.

Figure caption modified to:

Occurrence of industrial sites in permafrost dominated regions of the Arctic (permafrost occurrence probability >50%). The permafrost model domain is delineated based on the Northern Hemisphere permafrost map (20) and the database of industrial sites is based on OpenStreetMap (OSM) and the 2019 Nordregio Atlas of Population, Society and Economy in the Arctic (APSEA). While the industrial sectors "Energy" and "Agriculture, Forestry and Other Land Use" account for the largest proportion of industrial sites among the clearly labeled data, more than 65% of the mapped industrial sites are not clearly labeled. This creates a large uncertainty in quantifying specific industrial sectors and highlights the need for improved databases on industrial activities in the Arctic.

This figure seems to show the same information as Figure S3. I don't really understand why both are needed. I appreciate the addition of the final sentence here in response to my initial review but I still find the positioning of the circles from left to right quite distracting. I would recommend replacing this figure with S3 which is immediately understandable and actually includes more information.

We have replaced the Figure accordingly and updated the color scheme

You should specify the timeframe here even if it also appears in the methods.
Done.

I think this phrase is needed to explain why there are sites outside the permafrost limits shown (which is made clear in the methods but not here).

Done - rephrased to:

This database contains 44 individual incidents located within the permafrost dominated region and 58 incidents outside this region (Fig. 5c). Although our database contains only a small subset of the actual contaminated sites in Russia, the recorded incidents provide clear evidence that the spatial relationship between industrial sites and contaminated sites also applies to the Russian permafrost dominated region (SI Appendix, Fig. S5).

In this context, is 50 km a linear distance as indicated, or is it 50 km² or 50 x 50 km?

Specified accordingly:

Note that the speckled appearance results from the regional clustering of industrial sites combined with the chosen bandwidth (50 x 50 km) of the gaussian density filter used for the point process models.

May need to change text to reflect permafrost model domain not «on permafrost»

We have changed the terminology regarding the spatial domain under investigation throughout the manuscript.

I think this is a statement that may be true for modelling but is not necessarily true in the field. Permafrost in the discontinuous zone can be stable even if it occupies only half of the landscape. Distinguishing between the two is important.

We agree with the reviewer and omit the terms continuous and discontinuous permafrost in the revised version because these complex relationships and spatial variabilities cannot be accounted for in our modeling. We have also changed the relevant paragraph to better explain the scope of our model results.

The model has a coarse spatial resolution (1 degree), and thawing of permafrost within grid cells is considered likely if simulations indicate the formation of a talik (defined here as a permanent unfrozen soil layer greater than 0.1 m thick above permafrost). The simulations predict that about 22% of the currently existing industrial sites (~1,000) and 20 ± 4% of the estimated contaminated sites (2,200-4,800) are located in a region within which the simulations indicate that permafrost degradation is possible under present climate conditions (2020). The simulations further indicate that 15% of these industrial and contaminated sites are located in grid cells where a change from stable (talik free in the model) to unstable permafrost (talik in the model) occurred between 1960 and 2000.

Figure S4 shows the period 1980-2020. It would be better to harmonize the text with the figure or vice versa.

Done. See paragraph above.

I think it can be assumed that where possible, infrastructure would have been located on permafrost free terrain. So I don't think these two considerations have the same weight and this should be explained better. It is certain that some of the industrial sites where degradation is predicted were unaffected by this because they were not constructed on permafrost.

We agree and have modified our statement accordingly to:

It is likely that many of the industrial sites in this zone were built at permafrost-free locations where this was possible or where permafrost soil had been actively removed prior to construction, so that further warming would not affect their stability. However, at some of these sites, it cannot be ruled out that permafrost and possibly ground ice may still be present and that further warming will lead to soil instability and the formation of new hydrologic pathways.

Not clear what «its» refers to – sentence needs to be rephrased.

Concerning sentence changed to:

Currently available data are insufficient to assess the likelihood of future contamination or the risk of mobilization of contaminants due to permafrost degradation.

I don't think «amortization» is the correct word here – meaning is unclear.

Done. Wording changed to:

Consideration of these financial risks likely modifies the cost-benefit calculations for industrial activities in the Arctic.

Please specify if this is a distance (a radius?) or an area 1 km²?

Done.

To avoid double counting and resolve geometry conflicts between both datasets, we define buffer areas (1 km distance) around all features as overlap merging criteria.

This is an important assumption and deserves some additional justification. This should include the fact that the NHPM is at a scale of 1 km² while contaminated sites are generally much smaller than this and so may not be well represented in some cases.

We agree and thus have added the following more detailed explanation and classification:

Hence, sites in a grid cell characterized by a modeled permafrost occurrence probability greater than 50% were kept within the database, resulting in a total of 5,234 entries. Due to the spatial resolution of the NHPM (1 km), the occurrence of permafrost cannot be clearly assessed at the point locations of the industrial and contaminated sites. We therefore emphasize that we consider sites that are located in regions that are dominated by permafrost, but some of the actual sites may be permafrost free.

Supplementary material:

Please change the right-hand scale to start at zero in order to show that substantial amounts of crude oil continue to originate in Alaska.

Done.

Not clear if the frequency data are for all of Alaska or only for the modelled permafrost domain. I recommend replacing figure 4 with this figure.

Figure caption changed and Figure replaced.

Figure 4: Most frequently occurring toxic substances at contaminated sites in the permafrost dominated regions of Alaska (based on 20) as reported by the Contaminated Sites Program (CSP).

Better not to introduce a new term here, but if preferred, domain could be used throughout when referring to the permafrost region.

Done.

Figure S4: The observed and modeled point intensities of contaminated sites λ with the point density of industrial sites ρ for the permafrost dominated regions of Alaska and Canada.

Units are needed. Better to label the axes directly.

Units added.

References:

Hjort, J. et al. Degrading permafrost puts Arctic infrastructure at risk by mid-century. *Nature communications*, 9(1), 1-9 (2018).

Hugelius, G. et al. The Northern Circumpolar Soil Carbon Database: spatially distributed datasets of soil coverage and soil carbon storage in the northern permafrost regions. *Earth System Science Data* 5(1), 3-13 (2013).

Langer, M. et al. The evolution of Arctic permafrost over the last three centuries. *EGUsphere [preprint]*, 1-27 (2022).

Liew, M. et al. Understanding Effects of Permafrost Degradation and Coastal Erosion on Civil Infrastructure in Arctic Coastal Villages: A Community Survey and Knowledge Co-Production. *Journal of Marine Science and Engineering*, 10(3), 422 (2022).

Ran, Y.. Permafrost degradation increases risk and large future costs of infrastructure on the Third Pole. *Communications Earth & Environment*, 3(1), 1-10 (2022).

Suter, L.. Assessment of the cost of climate change impacts on critical infrastructure in the circumpolar Arctic. *Polar Geography*, 42(4), 267-286 (2019).